

# Accelerating Monte Carlo event generation – rejection sampling using neural network event-weight estimates

**Katharina Danziger[1], Timo Janßen[2], Steffen Schumann[2] and Frank Siegert[1]**

**1** Institut für Kern- und Teilchenphysik, TU Dresden, Dresden, Germany
**2** Institut für Theoretische Physik, Georg-August-Universität Göttingen, Göttingen, Germany

## Abstract

The generation of unit-weight events for complex scattering processes presents a severe challenge to modern Monte Carlo event generators. Even when using sophisticated phase-space sampling techniques adapted to the underlying transition matrix elements, the efficiency for generating unit-weight events from weighted samples can become a limiting factor in practical applications. Here we present a novel two-staged unweighting procedure that makes use of a neural-network surrogate for the full event weight. The algorithm can significantly accelerate the unweighting process, while it still guarantees unbiased sampling from the correct target distribution. We apply, validate and benchmark the new approach in high-multiplicity LHC production processes, including $Z/W+4$ jets and $t\bar{t}+3$ jets, where we find speed-up factors up to ten.

# 1 Introduction

Multi-purpose Monte Carlo event generators such as HERWIG [1,2], PYTHIA [3,4] or SHERPA [5, 6], are indispensable tools for the analysis and interpretation of high-energy particle-collision experiments, *e.g.* at the Large Hadron Collider (LHC). They encapsulate our present-day understanding of the fundamental laws of nature, and provide means to simulate individual scattering events in a fully exclusive manner. With such virtual collisions we can quantify expected event yields and predict detailed final-state properties for in principle arbitrary scattering processes.

The central and often the computationally most expensive element of event simulations is a hard-scattering process – addressing the highest momentum-transfer interactions – that gets described by transition matrix elements evaluated in fixed-order perturbation theory. Given the enormous collision energies and impressive luminosities achieved at the LHC, paired with the excellent performance of the experiments, the need to provide evaluations of higher multiplicity hard-scattering processes is steadily growing. In view of the upcoming HL-LHC this becomes an even more pressing problem, requiring *much faster* event generation in order to match the expected event yields with the projected computing resources [7,8]. The underlying matrix-elements are calculated by dedicated matrix-element generators. Widely used tree-level tools such as ALPGEN [9], AMEGIC [10], COMIX [11], MADGRAPH [12] and WHIZARD [13] automatically construct tree-level amplitudes, but also provide efficient means to generate momentum configurations for the initial- and final-state particles taking part in the hard scattering. Furthermore, there exist dedicated tools for the construction and evaluation of one-loop amplitudes in QCD and the electroweak coupling, *e.g.* MADLOOP [14,15], MCFM [16,17], NJET [18], OPENLOOPS [19,20], POWHEGBOX [21], and RECOLA [22,23]. These tools can be used to compile fixed-order partonic cross-section computations and to probabilistically generate parton-level events. When incorporated into or interfaced to a multi-purpose event generator they provide the momentum-space partonic scattering events that get dressed by QCD parton showers, if applicable supplemented by an underlying event simulation, and finally transitioned to fully exclusive hadron-level final states by invoking a hadronisation model [24].

An efficient sampling of the final-state phase space is particularly crucial for complex scattering processes, where a single evaluation of the matrix element can take $\mathcal{O}(1s)$ [25]. Especially for experimental applications, *i.e.* the actual generation of pseudo data, including a simulation of the detector response, see *e.g.* refs. [26–28], unit-weight event samples are required, that are conventionally obtained from weighted events via *rejection sampling*. The resulting unit-weight events are unbiased random samples of fully uncorrelated probes of the target distribution given by the squared transition matrix element. They appear with frequencies that we would expect in a corresponding experiment. Although information about the target is lost in the unweighting step, the expensive detector simulation or other post processing of many events with minuscule weight gets avoided.

In modern matrix-element generators importance-sampling techniques are used, that account and possibly adapt [29] to the modal structures of the target, thereby employing knowledge about the propagator and spin structures of a given process [30]. These methods aim to reduce the inherent variance of the weight distribution of weighted event samples, and in turn also improve the unweighting efficiencies.

There have recently been a number of different strands of research to make optimal use of event-weight information, and, largely driven by algorithmic opportunities provided by novel machine-learning (ML) techniques, to optimise phase-space sampling and also event unweighting. On-the-fly reweighting methods are meanwhile routinely used to account for systematic uncertainties [31–33], or alternative physics models [34,35]. The use of MCMC techniques for exploring high-dimensional phase spaces has been studied in [36]. In [37]

the application of analysis-specific optimal sampling distributions was proposed, similar to methods of biasing event generation, *e.g.* to oversample tails of physical distributions [38]. A number of approaches to accelerate event generation based on (generative adversarial) neural networks have been presented [39–47][1]. An alternative and particularly attractive class of algorithms is based on *normalizing flows* [49–51], *i.e.* trainable bijectors parametrised by neural networks, see for instance [52–55], that can represent highly expressive importance-sampling maps [56,57]. Corresponding implementations and first applications of normalizing flows to Monte Carlo event generation in high-energy physics have been presented in [58–61]. Ref. [62] discussed the usage of GANs, trained on weighted Monte Carlo samples to produce unit-weight events. However, in order to guarantee the reproduction of the true target distribution, an additional post-processing step is needed. Possible solutions to this problem based on reweighting have been presented in [63, 64]. The application of Bayesian networks for event generation including the quantification of uncertainties has been presented in [65, 66].

We here propose an alternative approach to accelerate the unweighting procedure using ML methods. During the initial integration phase of a standard importance sampler we train a deep neural network to predict the event weight for given phase-space points. For complex processes, this surrogate is much cheaper to evaluate than the actual event weight. We therefore employ it in an initial rejection sampling. Only when the surrogate event weight gets accepted, we invoke a second unweighting step, where we account for the difference between the surrogate and the actual event weight. While a two-step unweighting procedure has been applied before [9], our combination with a neural-network surrogate gives it a new purpose. Given the neural network approximates the weight distribution reasonably well, we can significantly reduce the number of evaluations of the computationally expensive target function. Our approach easily generalises to non-positive targets and is thus suitable also for unweighted-event generation beyond the leading order in perturbation theory. We have implemented, validated and benchmarked the method in the SHERPA event-generator framework and here present results for tree-level $Z/W+4$ jets and $t\bar{t}+3$ jets production at the LHC.

The paper is organised as follows, in Sec. 2 we briefly review the basics of Monte Carlo event generation and event unweighting in the canonical approach. We then introduce our novel unweighting procedure, exemplified for a simple toy example. In Sec. 3 we discuss the neural-network setup and the used training procedure to obtain a predictor for the weight of scattering events. In Sec. 4 we describe our implementation of the new method in the SHERPA framework and present exemplary results for high-multiplicity LHC production processes. We conclude and give an outlook in Sec. 5.

## 2 Phase-space sampling and event unweighting

For sake of simplicity, we begin by considering the generic integral

$$I = \int_{\Omega} f(u') \, \mathrm{d}u', \qquad (1)$$

with $f$ a positive-definite target distribution $f : \Omega \subset \mathbb{R}^d \to [0, \infty)$ defined over the unit hypercube $\Omega = [0, 1]^d$. The Monte Carlo estimate of the integral is given by

$$I \approx E_N = \frac{1}{N} \sum_{i=1}^{N} f(u_i) = \langle f \rangle, \qquad (2)$$

---

[1]A critical review on the application of Generative Adversarial Networks (GANs) in the context of event generation has been presented in [48].

where we assumed $N$ uniformly distributed random variables $u_i \in \Omega$. The random points $u_i$ are interpreted as individual *events* and $w_i \equiv f(u_i)$ is called the corresponding *event weight*[2]; the integral is thus estimated by the average of the event weights $\langle w \rangle_N$. The standard deviation of the integral estimate is given by

$$\sigma_N(f) = \sqrt{\frac{V_N(f)}{N}} = \sqrt{\frac{\langle f^2 \rangle - \langle f \rangle^2}{N}}, \tag{3}$$

with $V_N$ the corresponding variance. Variance-reduction techniques aim for a minimisation of $V_N$, *e.g.* by a remapping of the input random variables $u$ to a non-uniform distribution $v : \Omega \to \overline{\Omega}$, called *importance sampling* [67]. For the desired integral this results in

$$I = \int_\Omega \frac{f(u')}{g(u')} g(u') \, \mathrm{d}u' = \int_{\overline{\Omega}} \frac{f(u')}{g(u')} \bigg|_{u'=u'(v')} \mathrm{d}v' \quad \text{with} \quad g(u) = \left| \frac{\partial v(u)}{\partial u} \right|. \tag{4}$$

With suitably chosen probability density $g(u)$, the variance of the integrand can be significantly reduced. A prominent example widely used in particle physics is VEGAS [68]. Given the multimodal nature of high-energy scattering matrix elements, state-of-the-art generators employ adaptive multi-channel importance samplers [10, 11, 69, 70]. Thereby the probability density $g(u)$ is decomposed into a sum of $N_c$ *channels*, *i.e.*

$$g(u) = \sum_{j=1}^{N_c} \alpha_j g_j(u), \quad \text{with} \quad \sum_{j=1}^{N_c} \alpha_j = 1 \quad \text{and} \quad 0 \le \alpha_j \le 1, \tag{5}$$

yielding

$$I = \int_\Omega \frac{f(u')}{g(u')} \sum_{j=1}^{N_c} \alpha_j g_j(u') \, \mathrm{d}u' = \sum_{j=1}^{N_c} \alpha_j \int_{\overline{\Omega}} \frac{f(u')}{g(u')} \bigg|_{u'=u'(v'_j)} \mathrm{d}v'_j. \tag{6}$$

The channel weights $\alpha_j$ can thereby be adjusted dynamically such that the variance of the integral gets minimised [29].

To sample *unit-weight events* from the target function $f(u)$, typically a rejection sampling algorithm [71] is employed that utilises the maximal event weight in the integration volume, $w_{\max}$. A sample of $N^{\text{trials}}$ weighted events is thus converted into a set of $N \le N^{\text{trials}}$ unweighted events, where $N$ corresponds to the number of accepted events. The related unweighting efficiency for large $N$ is given by

$$\epsilon := \frac{N}{N^{\text{trials}}} = \frac{\langle w \rangle_{N^{\text{trials}}}}{w_{\max}}. \tag{7}$$

Its inverse determines the average number of target-function evaluations needed before an event is accepted with unit-weight.

An exact determination of $w_{\max}$ is often neither possible – given finite statistics – nor desirable in a numerical calculation that might exhibit a few points with spuriously large weights, as this would yield a prohibitively small unweighting efficiency. Instead, to avoid being dominated by such rare outliers, there are various possibilities to define a reduced maximum such that some "overweight" events with $w > w_{\max}$ are allowed and will be assigned a correction weight $\widetilde{w} = w/w_{\max}$, effectively leading to *partially unweighted* events[3]. Ref. [59] proposed a bootstrap method where the maximum is given by the median of $n$ determinations from

---

[2]In the following we drop the index $i$ as we are always referring to the generation of a single event.

[3]While the event weight $w$ is typically a dimensionful quantity, in unweighted events the weights $\widetilde{w}$ are considered dimensionless. To obtain the correct normalisation of a differential cross section, *e.g.* in a histogram, they need to be normalised to the *generated* inclusive cross section as reported by the event generator, $\widetilde{w}_i \to \widetilde{w}_i \cdot \frac{\sigma_{\text{gen}}}{\sum_j \widetilde{w}_j}$.

independent event batches. A more conventional approach would be the exclusion (from the maximum definition) of large-weight events with a certain quantile of the cross section[4]. In what follows we will make use of both techniques. The classical unweighting algorithm with overweight treatment for generating a single event is sketched in Alg. 1.

---

**Algorithm 1:** The classic rejection-sampling unweighting algorithm.

**while** *true* **do**
    generate phase-space point $u$;
    calculate exact event weight $w$;
    generate uniform random number $R \in [0,1)$;
    **if** $w > R \cdot w_{max}$ **then**
        **return** $u$ *and* $\widetilde{w} = \max(1, w/w_{max})$
    **end**
**end**

---

The application of variance-reduction methods will typically also lead to an improved unweighting efficiency $\epsilon$. In fact, an optimal sampler would directly produce event weights $w = \text{const}$, resulting in an unweighting efficiency of 100%. However, in realistic use cases this is never achieved. For high-multiplicity scattering processes unweighting efficiencies are instead often well below 1% [25, 59]. To systematically improve $\epsilon$ one needs to reduce $w_{\text{max}}$. The FOAM [72, 73] algorithm attempts to achieve this and aims for an optimised unweighting efficiency by gaining control over the maximal event weight.

## 2.1 A novel unweighting procedure

We here propose an alternative method aiming for a reduction of the computational resources needed to produce unweighted events that follow the desired target distribution. This can be achieved through a light-weight surrogate for the full event-weight calculation that enters a two-staged rejection-sampling algorithm. Given such a local surrogate $s$ for the true event weight $w$, that can for example be obtained from a well-trained neural-network predictor, *cf.* Sec. 3, we can use this surrogate in an initial rejection sampling against the maximal event weight $w_{\text{max}}$. However, to ultimately sample from the correct distribution, we need to account for the mismatch between the estimated and the actual event weight. This is accomplished with a correction factor $x = w/s$. This factor could be applied as an additional weight to accepted events, or a second rejection sampling step can be added to unweight this against the (predetermined) maximum, $x_{\text{max}}$, see below. The resulting unweighting algorithm for generating a single unit-weight event is sketched in Alg. 2 and explained in more detail in the following.

For a fast surrogate perfectly reproducing the exact weights, *i.e.* $x = 1$, the potential for saving resources is maximal, even though the unweighting efficiency obtained with the standard approach is not altered. This is the case, because for all trial configurations failing the first step only the surrogate gets evaluated, while the full weight is computed for accepted events only. However, in practice this is not realistic and the $x$ will vary around unity. Note, we do not require the approximation $s$ to overestimate $w$, and thus will also face values $x > 1$.

The appearance of non-unit relative weights $x$ makes a second unweighting step convenient. To this end, we need to predetermine the maximum $x_{\text{max}}$, against which to perform the additional rejection-sampling. Again, to avoid being dominated by rare outliers, we reduce $x_{\text{max}}$ in a controlled way by either excluding a certain quantile of the largest weights or using

---

[4]This is also the default in SHERPA for the standard rejection-sampling method.

---

**Algorithm 2:** Two-stage rejection-sampling unweighting algorithm using an event-wise weight estimate.

> **while** *true* **do**
>> generate phase-space point $u$;
>> calculate approximate event weight $s$;
>> generate uniform random number $R_1 \in [0,1)$;
>> # first unweighting step
>> **if** $s > R_1 \cdot w_{max}$ **then**
>>> calculate exact event weight $w$;
>>> determine ratio $x = w/s$;
>>> generate uniform random number $R_2 \in [0,1)$;
>>> # second unweighting step
>>> **if** $x > R_2 \cdot x_{max}$ **then**
>>>> **return** $u$ *and* $\widetilde{w} = \max(1, s/w_{max}) \cdot \max(1, x/x_{max})$
>>> **end**
>> **end**
> **end**

---

the median from several independent $x_{\max}$ determinations. We correct for the mismatch with the overweight $x/x_{\max}$ when $x > x_{\max}$. The final weight for an accepted event $u$ is then given by

$$\widetilde{w} = \max\left(1, \frac{s}{w_{\max}}\right) \cdot \max\left(1, \frac{x}{x_{\max}}\right). \tag{8}$$

As consequence of this residual weight, one might need to generate more events using the surrogate approach to achieve the same statistical accuracy as in standard unweighting. To account for this, we use the Kish effective sample size $N_{\text{eff}}$ [74] in the following,

$$N_{\text{eff}} := \frac{\left(\sum_i \widetilde{w}\right)^2}{\sum_i \widetilde{w}^2} = \alpha N, \tag{9}$$

where the sums run over all $N$ events passing the second unweighting and we introduced the proportionality factor $\alpha \leq 1$. The statistical accuracy of the sample is given by $1/\sqrt{N_{\text{eff}}}$. Only when using the true maximal weight $x_{\max}$, the effective sample size equals $N$, corresponding to $\alpha = 1$.

We can now introduce the effective gain factor $f_{\text{eff}}$ of the described two-staged unweighting procedure:

$$\begin{aligned} f_{\text{eff}} &:= \frac{T_{\text{standard}}}{T_{\text{surrogate}}} \\ &= \frac{N_{\text{eff}} \cdot \frac{\langle t_{\text{full}} \rangle}{\epsilon_{\text{full}}}}{N \cdot \left( \frac{\langle t_{\text{surr}} \rangle}{\epsilon_{\text{1st,surr}} \epsilon_{\text{2nd,surr}}} + \frac{\langle t_{\text{full}} \rangle}{\epsilon_{\text{2nd,surr}}} \right)} \\ &= \alpha \cdot \frac{1}{\frac{\langle t_{\text{surr}} \rangle}{\langle t_{\text{full}} \rangle} \cdot \frac{\epsilon_{\text{full}}}{\epsilon_{\text{1st,surr}} \epsilon_{\text{2nd,surr}}} + \frac{\epsilon_{\text{full}}}{\epsilon_{\text{2nd,surr}}}}. \end{aligned} \tag{10}$$

It accounts for all timing, efficiency, and statistical differences in the proposed event generation with Alg. 2 compared to standard (partially) unweighted event generation with Alg. 1. Here

$\langle t_{\text{full}} \rangle$ and $\langle t_{\text{surr}} \rangle$ denote the average evaluation times of the full weight and the surrogate, respectively. The quoted unweighting efficiencies are given by

$$\epsilon_{\text{full}} := \frac{N}{N_{\text{full}}^{\text{trials}}}, \quad \epsilon_{\text{1st,surr}} := \frac{N_{\text{2nd,surr}}^{\text{trials}}}{N_{\text{1st,surr}}^{\text{trials}}} \quad \text{and} \quad \epsilon_{\text{2nd,surr}} := \frac{N}{N_{\text{2nd,surr}}^{\text{trials}}}, \quad (11)$$

where the $N_{\text{step}}^{\text{trials}}$ denote the number of trials used in the respective unweighting step. We point out that phase-space cuts are applied before unweighting and therefore events rejected due to cuts do not count towards the number of trials here.

Significant speed gains can be expected if the standard unweighting efficiency $\epsilon_{\text{full}}$ is rather low and the surrogate approximates the true weights well, *i.e.* $\epsilon_{\text{1st,surr}} \approx \epsilon_{\text{full}}$ and $\epsilon_{\text{2nd,surr}} \approx 1$, while still being significantly faster, *i.e.* $\langle t_{\text{surr}} \rangle \ll \langle t_{\text{full}} \rangle$.

Note, the gain factor $f_{\text{eff}}$ has to be understood as an *upper bound* of a potential CPU time saving in an overall budget, as it does not apply to other stages of the event generation like parton showering and, more importantly, also not to post-processing steps like a detector simulation.

## 2.2 Generalisation to non-positive event weights

The above described unweighting method can easily be extended to the case of non-positive event weights. These naturally appear in higher-order perturbative calculations based on local subtraction methods such as Catani–Seymour [75] or Frixione–Kunszt–Signer [76] subtraction for next-to-leading-order (NLO) QCD calculations. In approaches matching and merging NLO matrix elements with QCD parton showers negative-weight events can resolve potential double counting of hard real-emission contributions and shower emissions off Born-like configurations, see for instance [77,78]. However, the appearance of such negative weights reduces the statistical significance of a fixed-size event sample as possibly large cancellations take place. In corresponding unweighted samples events contribute with weights ±1. The generalisation of the standard unweighting algorithm allowing for negative weights is given in Alg. 3. We thereby make use of a single maximal weight $w_{\text{max}} = |w|_{\text{max}} > 0$ in the rejection sampling, that is determined by the largest weight modulus observed in an initial exploration run[5].

---

**Algorithm 3:** Standard rejection-sampling unweighting algorithm allowing for negative-weight events.

**while** *true* **do**
    generate phase-space point $u$;
    calculate exact event weight $w$;
    generate uniform random number $R \in [0, 1)$;
    **if** $|w| > R \cdot w_{max}$ **then**
        | **return** $u$ *and* $\widetilde{w} = sgn(w) \cdot \max(1, |w|/w_{max})$
    **end**
**end**

---

This can be extended to our two-staged unweighting approach, using a surrogate for the full event weight that can now also become negative, *cf.* Alg. 4. We still employ a single maximal weight modulus in the first rejection step, where correspondingly we have to use the modulus of the surrogate, *i.e.* $|s|$. Similarly, for the second rejection sampling we use the modulus of the estimate for the maximal ratio between the full and the surrogate weights. Note

---

[5]As before, we consider the reduction of the maximum, compensated for by partial over-weighting.

that the sign of the ratio $w/s$ is not unique, as the surrogate $s$ might sometimes get the sign of the true weight wrong. Accordingly, we have to use $x = |w/s|$ also in the (partial) overweighting. The absolute weight value of an accepted event is still given by Eq. (8), however, its sign is determined by $\text{sgn}(\widetilde{w}) = \text{sgn}(w)$.

---

**Algorithm 4:** Two-stage rejection-sampling algorithm, allowing for negative valued (surrogate) weights. We hereby assume $w_{\max} > 0$ and $x_{\max} > 0$ given by the respective maximal modulus determined in a pre-run.

---

**while** *true* **do**
 generate phase-space point $u$;
 calculate approximate event weight $s$;
 generate uniform random number $R_1 \in [0, 1)$;
 # first unweighting step with $w_{\mathbf{max}} > 0$
 **if** $|s| > R_1 \cdot w_{max}$ **then**
  calculate exact event weight $w$;
  determine ratio $x = |w/s|$;
  generate uniform random number $R_2 \in [0, 1)$;
  # second unweighting step with $x_{\mathbf{max}} > 0$
  **if** $x > R_2 \cdot x_{max}$ **then**
   **return** $u$ *and* $\widetilde{w} = sgn(w) \cdot \max(1, |s|/w_{max}) \cdot \max(1, x/x_{max})$
  **end**
 **end**
**end**

---

To illustrate and validate the proposed algorithm we consider a simple $1d$ example by sampling from the target distribution

$$f(u) = u^2 - 0.25, \quad \text{for } u \in [0, 1]. \tag{12}$$

As surrogate we here just use a piecewise constant function over $u \in [0, 1]$ given by

$$s(u) = -0.25\chi_{[0,0.2)}(u) - 0.15\chi_{[0.2,0.4)}(u) + 0.05\chi_{[0.4,0.6)}(u) + 0.25\chi_{[0.6,0.8)}(u) + 0.75\chi_{[0.8,1]}(u), \tag{13}$$

where

$$\chi_M(u) = \begin{cases} 1 : u \in M \\ 0 : u \notin M \end{cases}. \tag{14}$$

This encloses the cases that the surrogate over- or underestimates the target, as well as predicting its sign wrongly. In the left panel of Fig. 1 we compile the target distribution, the surrogate and their ratio. Furthermore, we mark the maximum used in the second unweighting step, *i.e.* $x_{\max} = 1.5$. This is chosen such that there are regions where $|f(u)/s(u)| > x_{\max}$, triggering the appearance of events with weight $|w| > 1$. In the right panel of Fig. 1 we present the distributions obtained from 500k events generated with the standard unweighting algorithm and the two-staged approach. Comparing to the true target distribution we see that both methods produce the desired density. To further confirm the proper treatment for those events where $x > x_{\max}$, we provide a close-up view of the region around $u = 0.6$. In the standard approach the unweighting efficiency is $\epsilon_{\text{full}} = 0.33$, requiring $N_{\text{full}}^{\text{trials}} \approx 1.5\text{M}$ calls of the target function to generate 500k unit-weight events. In contrast, with the given surrogate and the choice of $x_{\max}$ we obtain $\epsilon_{\text{1st,surr}} = 0.39$ and $\epsilon_{\text{2nd,surr}} = 0.58$, corresponding to $N_{\text{surr}}^{\text{trials}} \approx 2.2\text{M}$. However, for the given event sample we only had to evaluate the target $N_{\text{full}}^{\text{trials}} \approx 875\text{k}$ times.

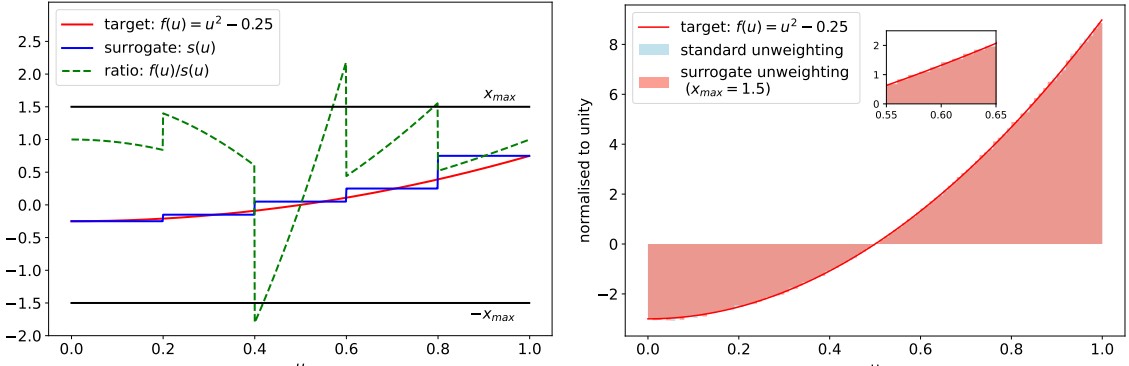

Figure 1: One-dimensional toy example for applying the standard unweighting algorithm and the two-staged surrogate method to a non-positive target function. The left panel shows the target (red) and the employed surrogate (blue), given by Eq. (13), as well as their ratio (green dashed). Indicated is the (capped) maximal ratio $x_{\text{max}}$ used in the second unweighting step. The right panel contains the comparison of the distributions of generated events with the true target.

## 3 Machine learning event weights

The calculation of transition matrix elements for complicated scattering processes, in particular when considering higher-order corrections, becomes computationally very expensive. In applications that require a large number of repeated evaluations this poses a severe bottleneck. The generation of unweighted events considered here is only one such example, others include the fitting of parton density functions (PDFs), or scans over large parameter spaces in searches for New Physics, *i.e.* corresponding limit-setting procedures.

For the fast evaluation of fixed-order differential cross sections needed in the determination of PDFs interpolation grids such as APPLGRID [79], FASTNLO [80], and PINEAPPL [81] are widely used, and there exist tools for their largely automated construction [82,83]. To facilitate and accelerate analyses searching for New Physics, there have recently been efforts to use deep-learning techniques for the regression of cross-section integrals [84–86]. Very recently also the approximation of scattering matrix elements rather than integrated cross sections through neural networks has been addressed by several groups [87–90]. These approaches suggest that high-quality surrogates for full scattering matrix elements are feasible, offering potential for significant speed-ups in the event-generation process when applied within the unweighting framework described above.

### 3.1 Neural-network based matrix-element emulation

For a first application of the surrogate-based unweighting, we introduce a custom ML model, which learns and predicts the complete weight of partonic scattering events. This combines the squared matrix element and the phase-space weight, the latter including Jacobian factors $J_{\Phi_n}$ from variable mappings of the Lorentz invariant phase-space element $\Phi_n$ used by the underlying integrator. For a given $2 \to n$ parton-level process our surrogate $s(p_a, p_b, p_1, \ldots, p_n)$ thus approximates the following part of the fully differential cross section:

$$\mathrm{d}\sigma_{ab\to n}\big|_{p_a, p_b, \{p_i\}} = \underbrace{f_a(x_a, \mu_F)\, f_b(x_b, \mu_F)\big|\mathcal{M}_{ab\to n}\big|^2 \big|J_{\Phi_n}\big|}_{\approx s}\, \mathrm{d}x_a\, \mathrm{d}x_b\, \mathrm{d}\Phi_n\big|_{p_a, p_b, \{p_i\}}. \qquad (15)$$

Here $f_{a/b}$ denotes the PDF for the incoming parton $a/b$ with momentum fraction $x_{a/b}$,

evaluated at factorisation scale $\mu_F$. Note, the PDF contribution could also be factored out of the surrogate and evaluated exactly on an event-wise basis but we here decided to include it. The external particle momenta satisfy four-momentum conservation and on-shell conditions:

$$p_a + p_b = \sum_{i=1}^{n} p_i, \ \ p_{a/b}^2 = 0, \ \text{and} \ p_i^2 = m_i^2 \quad (\forall\, i = 1, \ldots, n). \tag{16}$$

Accordingly, the dimensionality of the physical phase space is $d = 3n - 4 + 2$.

When comparing Eq. (15) to the first identity in the multi-channel integral given by Eq. (6), we identify the phase-space element $\mathrm{d}x_a\,\mathrm{d}x_b\,d\Phi_n$ in momentum space with the differential $du'$ multiplied by the multi-channel density $\sum_j \alpha_j g_j(u')$. The Jacobian factor $|J_{\Phi_n}|$ corresponds to $1/g(u')$. Our NN thus has to approximate the ratio $f(u')/g(u')$, that is obviously dependent on the total importance sampling density $g$, but not on the very channel used to produce the phase-space point, see also Ref. [69].

Alternatively to Eq. (15) one could approximate the squared matrix element only, *i.e.* $s' \approx \left|\mathcal{M}_{ab \to n}\right|^2$, and fully calculate the Jacobian factors for each phase-space point. Due to its factorised nature, this approach would in fact be easier to implement. However, it suffers from the significant costs of evaluating the phase-space weight for multi-leg processes, which can sometimes even rival the evaluation cost of the matrix element. Furthermore, the combination of Jacobian factors and matrix elements often yields a smoother function over phase space. We thus only consider the approach of replacing the combined matrix-element and phase-space weight with a fast surrogate here.

Various test cases for surrogate models were considered in the course of this work, including (boosted) decision trees, random forests and neural networks. While being faster[6], random forests and (boosted) decision trees yield a poorer prediction accuracy, rendering them inadequate for an application in the surrogate-based unweighting [91]. Thus, only neural networks are discussed further in the following.

Given the specific role of the surrogate in the proposed unweighting procedure, we seek for light-weight network architectures, flexible enough to approximate the weight of high-multiplicity scattering events well, and fast to evaluate. To this end we employ rather simple multi-layer feedforward fully connected neural networks (NN).

As input-layer variables we use the three-momentum components of the initial- and final-state particles[7], *i.e.* $3n + 2$ inputs. In general, any set of variables that has an injective mapping to the phase-space point could be used, even with different dimensionality if adding or removing variables.

One might alternatively consider a particular set of input variables, namely the random numbers $v_i$ from the phase-space sampling, which have been mapped into momenta as described by Eqs. (4) and (6). While this is straightforward for simple sampling methods, it becomes more tricky for multi-channel samplers. Here, the mapping between random numbers and phase-space point is not unique, but depends on the randomly chosen channel $j = 1 \ldots N_c$. To remedy this situation, one could either train a separate NN for each phase-space channel $j$, or one could add the channel number $j$ (or the random number determining it) as another input variable. We postpone a study of these possibilities to future works.

The single output variable of our NN corresponds to the real-valued event weight. The network is further defined by the number of hidden layers and the set of nodes per layer as detailed in Table 1. As output activation function for the network nodes we use the Rectified

---

[6]The prediction speed of the machine-learning models depends on their architecture. One can construct simple neural networks which are able to predict faster than a very deep decision tree. However, the accuracy and ability to generalise may decrease with simpler topologies.

[7]Note, we here assume the initial-state momenta of partonic scattering events to be collinear with the incoming beams, *i.e.* along the $\pm z$-axis, such that their $x$ and $y$ components vanish.

Linear Unit (ReLU) [92]. We use HE weight initialisation [93] and train the NN with the ADAM optimiser [94].

The practical implementation of NN training in the SHERPA framework and the interface for (general) surrogate models for application in event unweighting will be detailed in Sec. 4. In the remainder of this section, however, details on the hyperparameters of our NN and the training procedure are given. The NN performance is first studied for the example process $gg \to e^-e^+ggd\bar{d}$. We used this channel as a test bed for investigations on the NN performance in terms of timing and the quality of the event-weight predictions as a function of the hyperparameters. Being primarily interested in a conceptual proof-of-concept and an initial estimation of the method's potential to save resources in event unweighting, we do not attempt to systematically optimise the NN setup. Furthermore, while in principle different scattering processes might get better approximated by a different NN architecture, we will employ the hyperparameter set found in the following example also in our other applications presented in Sec. 4.

## 3.2   An example: $gg \to e^-e^+ggd\bar{d}$

We consider the partonic channel $gg \to e^-e^+ggd\bar{d}$ at the leading order, *i.e.* $\mathcal{O}(\alpha^2\alpha_s^4)$, that represents a tree-level contribution to $Z+4$ jets production at the LHC. Correspondingly, the input-parameter space for the NN here is 20-dimensional. The fiducial phase space used in the training and for the predictions is constrained by requiring a dilepton invariant mass $m_{e^-e^+} > 66$ GeV and four anti-$k_t$ jets [95] with radius parameter $R = 0.4$ and $p_{T,j} > 20$ GeV. We consider a proton–proton centre-of-mass energy of $\sqrt{s} = 13$ TeV, and use the NNPDF-3.0 NNLO PDF set [96]. As matrix-element and phase-space generator we employ AMEGIC [10] in the framework of SHERPA-2.2.

Our NN has four hidden layers with 128 nodes each. The training dataset consists of 1M events generated with SHERPA after the optimisation phase of the AMEGIC integrator. We split the dataset such that 80% of the events are used for training and 20% for validation. In order to normalise the input features, we scale the momenta to the range $[-1, 1]$ using min-max normalisation with the min-max values given by $\pm\sqrt{s}/2$. As the values of the weights can span several orders of magnitude, we take the logarithm of the weights in order to avoid numerical problems. The NN model is fitted to the data by minimising the mean squared error (MSE) loss using the ADAM optimiser with a learning rate of $10^{-3}$. We use a batch size of 1000 and train in epochs containing all training points in random order. Early stopping is used to end the training when the validation loss does not decrease for 30 epochs and save the model with the lowest validation loss. Like for the training we also use the MSE loss for validation. Fig. 2 shows the convergence behaviour of our model. One can see that the loss decreases fairly smoothly and that the variations between different initialisations of the model are small.

To test the quality of our trained NN surrogate $s$ for the true event weights $w$ we present in Fig. 3a the resulting distribution of $x = w/s$ for 1M phase-space points generated with SHERPA. The $x$-distribution is centred around $x = 1$, rather symmetric, and falls off quite steeply. This confirms that the chosen NN is indeed suitable for a prediction of the event weight. Still we observe that the tails of the distribution stretch beyond $|\log_{10}(x)| > 4$, meaning the NN sometimes severely over- or underestimates the true weight. In particular the largest $x$-values will affect the performance of the unweighting algorithm proposed in Sec. 2.1, as they determine the maximum $x_{\max}$ against which to perform the second rejection sampling. Fig. 3b shows that the largest and smallest values of $x$ correspond to small values of $w$. As opposed to this, the NN approximation is much better for higher values of $w$ as can be recognized by the smaller spread of the $x$-values. This behaviour can be expected given the MSE loss function used for the training of the NN. While Fig. 3b shows that the largest relative deviations can be found for small values of $w$, the absolute deviations in that region are actually small. The MSE

Table 1: Summary of hyperparameters specifying the employed feedforward NN architecture and the means of training.

| NN Hyperparameter | Value |
| --- | --- |
| Hidden Layers | 4 |
| Nodes per Layer | 128 |
| Activation Function | ReLU |
| Loss Function | MSE |
| Optimiser | ADAM |
| Learning Rate | $10^{-3}$ |
| Batch Size | 1000 |

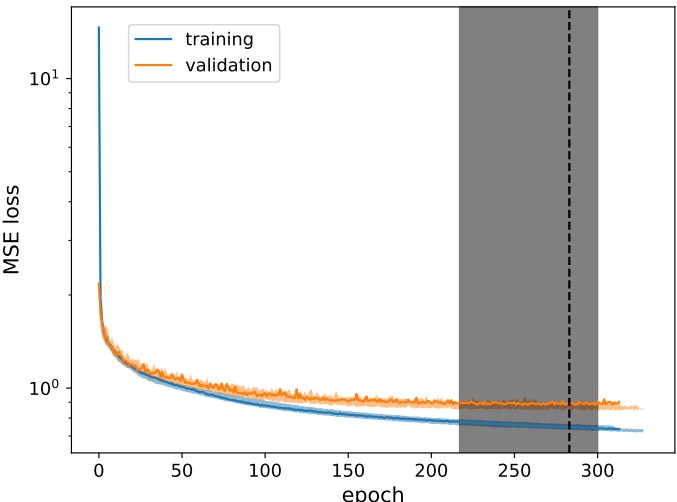

Figure 2: Training (blue) and validation (orange) MSE loss of the best performing NN during training. The dashed line illustrates the stopping point due to early stopping. The coloured bands show the variations from ten independently trained initialisations of the same model.

loss function penalizes absolute deviations at larger $w$-values more than at smaller $w$-values which leads to larger relative deviations for small values of $w$.

As described already in the context of the first unweighting step in Alg. 1, we can use maximum-reduction techniques also for $x_{\mathrm{max}}$ in the second rejection sampling. These will reduce the sensitivity to the tail of the weight distribution and in particular rare outliers by using a reduced maximum, again at the price of a partial overweighting of events. In our performance study in Sec. 4 we will employ two reduction techniques. The first being the *quantile reduction method*, where we define $x_{\mathrm{max}}^{\mathrm{p.m.}}$ such that the remaining overweights contribute at most $1‰$ to the total cross section $\sigma$. We consider an event sample of $N = 1\mathrm{M}$ events with weights $\{w_i\}$. For reference, in the standard unweighting method we can determine $w_{\mathrm{max}}^{\mathrm{p.m.}}$ by sorting the sequence of weights $\{w_i\}$ such that $w_i \leq w_{i+1}$ and requiring that

$$w_{\mathrm{max}}^{\mathrm{p.m.}} := \min\left( w_j \,\middle|\, \sum_{i=j+1}^{N} w_i < 0.001 \cdot \sum_{i=1}^{N} w_i \right). \tag{17}$$

The equivalent procedure for our two-stage unweighting method is to calculate the values of

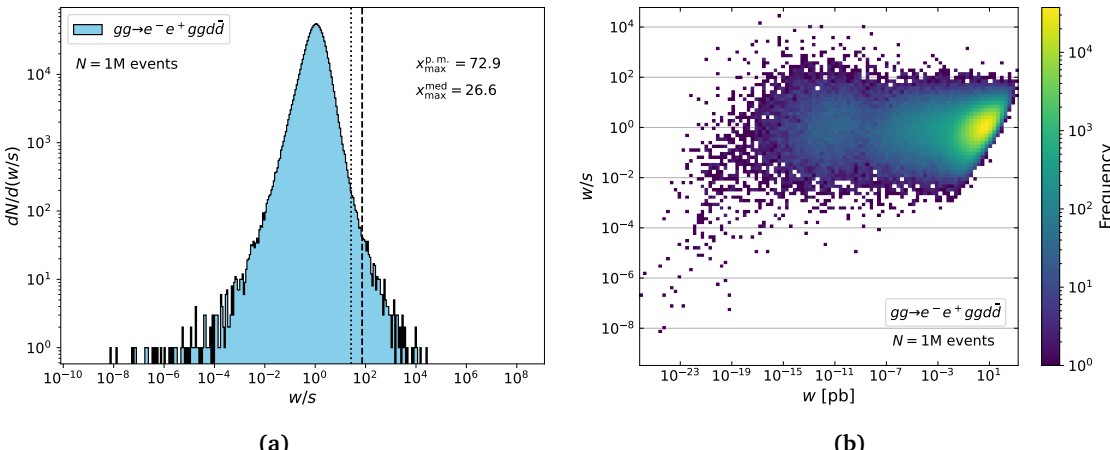

**(a)**                **(b)**

Figure 3: Distribution of weights using 1M test points generated with SHERPA for the process $gg \to e^-e^+ggd\bar{d}$ in proton–proton collisions at $\sqrt{s} = 13$ TeV. **(a)** One-dimensional histogram of the ratio $x = \frac{w}{s}$. The two vertical lines indicate the values of the reduced weight maxima $x_{\max}^{\text{p.m.}}$ (dashed) and $x_{\max}^{\text{med}}$ (dotted). **(b)** Two-dimensional histogram showing the relationship between the ratio $x = \frac{w}{s}$ and the true event weight $w$.

$s$ and $x$ for all events and to sort the sequence $\{x_i\}$ such that $x_i \leq x_{i+1}$ and to use the same order for the $\{s_i\}$. The reduced maximum is then defined as

$$x_{\max}^{\text{p.m.}} := \min\left( x_j \;\middle|\; \sum_{i=j+1}^{N} x_i s_i < 0.001 \cdot \sum_{i=1}^{N} x_i s_i \right). \tag{18}$$

As a somewhat more aggressive alternative we introduce the *median reduction method*. Here we consider $N_{\text{1st,surr}}^{\text{trials}} = 1$M trial points for which we perform the first unweighting $n = 50$ times with different random seeds. For the accepted events in each iteration we determine $x_{\max}$. From the final set of maxima we then determine the median $x_{\max}^{\text{med}}$, *i.e.*

$$x_{\max}^{\text{med}} := \text{med}\left( \{x_{\max}^{i}\} \right). \tag{19}$$

For our example process the resulting values of $x_{\max}^{\text{p.m.}}$ and $x_{\max}^{\text{med}}$ are illustrated by the vertical dashed and dotted line in Fig. 3a, respectively. In this specific example we obtain $x_{\max}^{\text{p.m.}} \approx 73$ and $x_{\max}^{\text{med}} \approx 27$, which corresponds to a reduction of $x_{\max}^{\text{p.m.}}$ by about two orders of magnitude with respect to the naive maximum, and an additional factor of three when using the median approach.

We close this section with a comment on the timings for the evaluation of the matrix element and the NN surrogate for a single phase-space point. On average the evaluation of the full event weight for the $gg \to e^-e^+ggd\bar{d}$ process from AMEGIC takes about 85 ms[8]. In contrast, for the NN model this just takes 0.13 ms, which translates into a speed-up of

$$\frac{\langle t_{\text{full}} \rangle}{\langle t_{\text{surr}} \rangle} \approx 650. \tag{20}$$

---

[8]The quoted times correspond to the evaluation on a single core of an Intel® Xeon® Processor E5-2680 v3 @ 2.50GHz.

# 4  Surrogate-based unweighting: Implementation, Validation and Results

The event-weight estimator from Sec. 3.1 is optimally suited to be used as light-weight surrogate in the two-stage unweighting method presented in Sec. 2.1. In the following, we will briefly describe the implementation of the algorithm in the SHERPA generator framework. As a first application we will again consider the $gg \rightarrow e^- e^+ gg d\bar{d}$ process. We will then benchmark the method in a variety of partonic channels contributing to $W+4$ jets and $t\bar{t}+3$ jets production at the LHC and validate the obtained results.

## 4.1  Implementation in the SHERPA framework

The SHERPA framework embeds modules to automatically construct the transition matrix elements and suitable multi-channel integrators for in-principle arbitrary tree-level processes. To this end it has two matrix-element generators built-in, AMEGIC and COMIX. Our current implementation of the novel unweighting algorithm employs the AMEGIC generator.

In an initial optimisation phase the integrator is adapted to the specific process and fiducial phase space using the channel-weight optimisation described in [29]. During the integration phase the value of $w_{\max}$ is determined based on the quantile approach. We use the SHERPA default of letting overweighted events have a relative contribution of $1\text{‰}$ to the inclusive cross section. The optimised generator is then used to produce a sample of 2M weighted events. We use the first 1M events as training (80%) and validation (20%) data for our NN model.[9] For the NN implementation and training we use KERAS [97] with the TENSORFLOW [98] backend. The model parameters leading to the lowest validation loss are written out as an HDF5 [99] file. While KERAS is based on Python, SHERPA is written in C++. To use the KERAS model in SHERPA without having to rely on an interface we use the header-only library frugally-deep [100] which runs the model in prediction mode on a single CPU core.

The second 1M events are used to determine the $x_{\max}$ for the second unweighting using the per mille quantile or median approach. For the latter we consider $n = 50$ independent iterations over the data set. This procedure is repeated for ten independently trained NN models and we finally choose the one achieving the lowest $x_{\max}$ on the test dataset to be used in the following. The NN and the value of $x_{\max}$ then serve as inputs to SHERPA for subsequent event-generation runs. We use different events for the determination of $x_{\max}$ than for the training of the NN. If one were to use the same data set, $x_{\max}$ would likely be underestimated. With data not seen by the model during training, however, we get a much more reliable estimate.

For the performance analysis we log several quantities during the event generation. To determine the efficiencies, we count the numbers of trials for the first and second unweighting steps. Also, we measure the time it takes on average to evaluate the surrogate by taking the sum of user and system time spent in the respective parts of the code.

## 4.2  An example: $gg \rightarrow e^- e^+ gg d\bar{d}$

Before proceeding with the application of our novel unweighting approach to $W+4$ jets and $t\bar{t}+3$ jets production at the LHC, we examine its technical and physics performance in more detail for the example process of $gg \rightarrow e^- e^+ gg d\bar{d}$. This is the channel initially used to optimise the NN performance in terms of timing and accuracy, *cf.* Sec. 3.2.

---

[9]In a production implementation in the future, one could also perform the training on the same events that are generated during the integration phase after the integrator optimisation.

**Performance analysis**

The evaluation of the NN surrogate for a single phase-space point was found to be about 650-times faster than the full weight calculation with AMEGIC. In Fig. 3a we have presented the obtained distribution of $x = w/s$, where we also indicated the reduced maxima for the per mille quantile and the median approach, *i.e.* $x_{max}^{p.m.} \approx 73$ and $x_{max}^{med} \approx 27$, respectively. Using the trained NN and each of these maxima, we generate from scratch 100k events with our surrogate unweighting algorithm. In Tab. 2 we summarise the obtained efficiency of the default single-stage unweighting, $\epsilon_{full}$, the efficiencies of the first and second rejection-sampling step in the surrogate unweighting, as well as the $\alpha$-parameter that determines the effective sample size, *cf.* Eq. (9), for the two maximum-reduction methods[10]. Lastly, we give the resulting gain factors $f_{eff}$, *cf.* Eq. (10).

Table 2: Sampling measures for the $gg \to e^- e^+ g g d \bar{d}$ partonic channel in $pp$ collisions at $\sqrt{s} = 13$ TeV. All efficiencies, the sample-size parameters and effective gain factors are determined in the generation of 100k unweighted events.

| process: $gg \to e^- e^+ g g d \bar{d}$ | | | | | | | | | |
|---|---|---|---|---|---|---|---|---|---|
| $\epsilon_{full}$ | $\epsilon_{1st,surr}$ | $x_{max}^{p.m.}$ | $\epsilon_{2nd,surr}^{p.m.}$ | $\alpha^{p.m.}$ | $f_{eff}^{p.m.}$ | $x_{max}^{med}$ | $\epsilon_{2nd,surr}^{med}$ | $\alpha^{med}$ | $f_{eff}^{med}$ |
| 8.8e−3 | 6.4e−3 | 72.9 | 1.9e−2 | 0.9982 | 1.73 | 26.6 | 5.1e−2 | 0.9962 | 4.69 |

Using the default unweighting algorithm, AMEGIC achieves an unweighting efficiency of about 0.9%. This in fact is quite remarkable, given that we consider a six-particle final state. When using the NN surrogate we obtain a similar performance, $\epsilon_{1st,surr} \approx 0.64\%$, and given the fast evaluation time for the surrogate this slightly lower efficiency barely affects the overall performance. More relevant is the second unweighting, for which we find efficiencies of $\epsilon_{2nd,surr}^{p.m.} = 1.9\%$ and $\epsilon_{2nd,surr}^{med} = 5.1\%$. Accordingly, when using the median-reduction technique, we need to evaluate the full weight roughly a factor 2.7 less often than for the quantile approach. For the considered process this almost directly transfers to the effective gain factors that yield $f_{eff}^{p.m.} = 1.73$ and $f_{eff}^{med} = 4.69$. These gains are a consequence of the speed of the surrogate evaluation, and its excellent approximation of the true weights, *i.e.* the very steep fall-off of the $x = w/s$ distribution. In fact, the effective sample size reduces only to 99.8 % and 99.6 % of a unit-weight sample, which will be negligible in practical applications.

The obtained $\alpha$ values close to unity reflect the fact that only few events retain non-unit weights $\widetilde{w}$ in the end, *cf.* Eq. (8). This is confirmed by Fig. 4 where we present the final event-weight distribution for the sample of 100k events generated using the more aggressively reduced maximum $x_{max}^{med}$ in the second unweighting step. Indeed, only a small fraction of events exhibits weights $\widetilde{w} > 1$. Furthermore, the overweights rarely exceed $\widetilde{w} = 3$ and the maximum we observe within this sample is $\widetilde{w} \approx 9$.

**Physics validation**

To prove that our algorithm indeed produces the correct target distribution we now move to the validation of differential cross sections. Figure 5 collects various physical observables comparing the predictions of SHERPA with and without the novel unweighting approach. For both methods we produced samples of 1M events at the parton level. Parton shower and hadronisation effects are disabled in these and the following simulations to increase the resolution and sensitivity to potential differences between the two approaches. These were analysed with

---

[10]Note, the $w_{max}$ used in the first unweighting is always reduced using the per mille quantile approach to keep the full and the surrogate approach comparable.

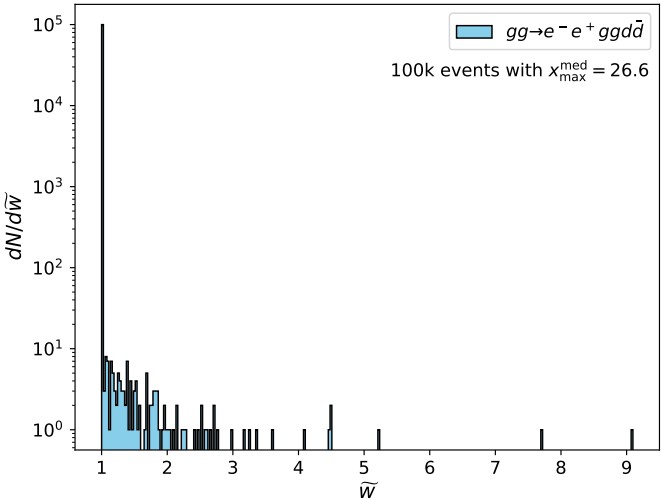

Figure 4: Final event weights $\widetilde{w}$ of 100k $gg \rightarrow e^- e^+ g g d \bar{d}$ events in proton–proton collisions at $\sqrt{s} = 13$ TeV generated using surrogate unweighting with $x_{\max}^{\mathrm{med}}$.

the RIVET3 toolkit [101] using the MC_ZINC and MC_ZJETS analyses. In panel (a) we show the dilepton invariant mass, (b) the dilepton rapidity distribution, (c) the $p_T$ of the jet with highest transverse momentum, and (d) the azimuthal distance between the two leading jets. For each plot we provide two sub-panels. In the first we depict the ratio of the predictions obtained from the surrogate approach and nominal SHERPA, where the errorbars indicate the bin-wise statistical uncertainty. The second panel displays directly the statistical compatibility of the two predictions measured in terms of standard deviations.

For all four observables we find full statistical agreement, which proves that the surrogate approach produces the correct target function. This also applies to the tail of the distributions. No significant increase in the statistical errors is observed for the surrogate-based prediction, which verifies the negligible reduction of $\alpha^{\mathrm{med}} = 0.9962$. Furthermore, there is no visible imprint of statistical fluctuations from the events that exceed the maximum in the second unweighting.

For the considered example process we can conclude that when using the surrogate unweighting approach we can generate samples of almost identical statistical accuracy that reproduce the exact physical distribution. Depending on the method used to reduce the maximum in the second unweighting step we find effective gain factors up to 4.7.

## 4.3 Results for LHC production processes

In this section we present results for processes contributing to $W+4$ jets and $t\bar{t}+3$ jets production at the LHC, providing further insight into the potential and limitations of the surrogate-unweighting method. Both final states receive contributions from a large number of partonic channels, from which we pick representatives here. Given the high final-state multiplicity, the large number of contributing Feynman diagrams, and the complexity in QCD colour space, these matrix elements are highly non-trivial functions over phase space and rather expensive to evaluate, such that we can expect gains from employing the surrogate method.

In the following we employ the same network architecture and training measures as described in Sec. 3.1 and used in the $Z+4$ jets example and apply them in each partonic channel separately. We do not attempt to specifically adjust and optimise the hyperparameters, though this could potentially further improve performance. As before, all setups are studied with SHERPA-2.2 for $pp$ collisions at $\sqrt{s} = 13$ TeV, using the NNPDF-3.0 NNLO PDF set and AMEGIC

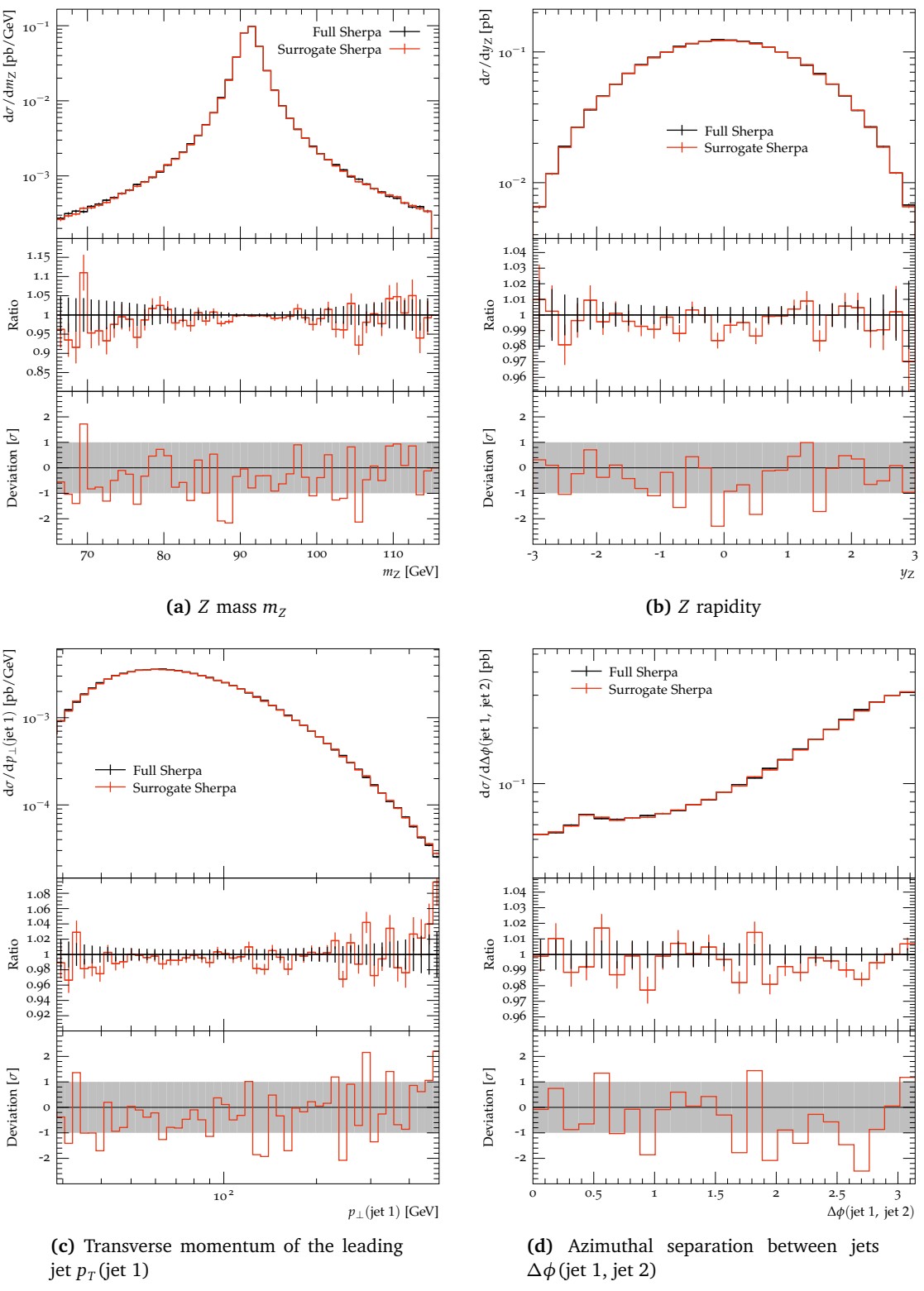

**(a)** $Z$ mass $m_Z$

**(b)** $Z$ rapidity

**(c)** Transverse momentum of the leading jet $p_T$ (jet 1)

**(d)** Azimuthal separation between jets $\Delta\phi$ (jet 1, jet 2)

Figure 5: Comparison of different differential distributions generated using SHERPA with (red) and without (black) an NN weight surrogate for the process $gg \to e^-e^+ggd\bar{d}$ in proton–proton collisions at $\sqrt{s} = 13$ TeV.

as matrix-element and phase-space generator.

Besides quantifying timing improvements, we scrutinise the physics description by validating observable distributions against the standard unweighting approach. It is worth mentioning that all presented timing improvements can likewise be translated into energy savings as no notable additional computing resources are needed for the new approach.

### 4.3.1 $W$+4 jets

We consider three partonic channels with varying numbers of external gluons that contribute to $W$+4 jets in proton–proton collisions. These are listed along with their respective tree-level production cross section in Tab. 3. While the dimensionality of the input-parameter space for the NN surrogate is identical to the $Z$+4 jets example, we now consider the charged-current weak interaction and different combinations of initial- and final-state partons.

Table 3: Selection of partonic channels contributing to $W$+4 jets production at the LHC and their corresponding leading-order production cross sections.

| process | cross section [pb] |
|---|---|
| $dg \to e^- \bar{\nu}_e g g g u$ | 24.5(2) |
| $dd \to e^- \bar{\nu}_e g g d u$ | 4.62(3) |
| $ud \to e^- \bar{\nu}_e d u u \bar{d}$ | 0.0572(3) |

The quoted cross sections correspond to a fiducial phase space requiring four anti-$k_t$ jets with $R = 0.4$ and $p_{T,j} > 20\,\text{GeV}$, and $m_{e^- \bar{\nu}_e} > 1$ GeV. Due to the high total production rate, $W$+4 jets final states constitute an important background to top-quark pair-production and many searches for new physics phenomena. From Tab. 3 we can infer that the cross sections of different partonic channels vary significantly. In particular processes with more external gluons dominate over quarks. An additional driver are the initial-state flavour PDFs. The larger the contribution of a partonic channel to the total $W$+4 jets cross section the more events it will contribute to an inclusive sample. Accordingly, it is desirable to speed up event generation in particular for the dominant production channels.

**Performance analysis**

In Tab. 4 we compile the performance measures for unweighted event generation separately for the three considered $W$+4 jets partonic channels. They are determined from samples of 100k events generated with the standard and the NN surrogate approach.

Notably, for all three processes the standard unweighting efficiency is lower than for the $Z$+4 jets channel. For the process with four external gluons the evaluation of the surrogate model is again more than 600 times faster than the full weight calculation. However, for the other two cases we achieve speed-up factors of 162 and 25 only. These lower gains originate from shorter evaluation times for the full weights of $20\,\text{ms}$ for $dd \to e^- \bar{\nu}_e g g d u$ and $3\,\text{ms}$ for $ud \to e^- \bar{\nu}_e d u u \bar{d}$, while the NN surrogate takes about $0.12\,\text{ms}$ for *each* channel. While the maxima $x_{\max}^{\text{med}}$ are all of a similar size as in the $Z$+4 jets case, the values for $x_{\max}^{\text{p.m.}}$ are significantly higher, ranging up to 1650 for $ud \to e^- \bar{\nu}_e d u u \bar{d}$. This suggests that the NN provides an inferior approximation of the weights for the processes and fiducial phase space considered here. To illustrate this we show in Fig. 6a the distribution of $x = w/s$ for 1M events for the process $dd \to e^- \bar{\nu}_e g g d u$. When comparing to Fig. 3a we indeed observe a broader distribution that exhibits more pronounced tails. The two vertical lines indicate the values of $x_{\max}^{\text{p.m.}}$ (dashed) and $x_{\max}^{\text{med}}$ (dotted). By comparing the relationship between $x$ and $w$ shown in Fig. 6b to the one shown in Fig 3b, we see that the spread of $x$-values is much broader overall. However,

Table 4: Performance measures for partonic channels contributing to $W+4$ jets production at the LHC.

| | $dg \to e^-\bar{\nu}_e gggu$ | $dd \to e^-\bar{\nu}_e ggdu$ | $ud \to e^-\bar{\nu}_e duu\bar{d}$ |
|---|---|---|---|
| $\epsilon_{\text{full}}$ | 1.4e−3 | 3.1e−4 | 3.6e−4 |
| $\epsilon_{\text{1st,surr}}$ | 7.1e−4 | 1.1e−4 | 1.3e−4 |
| $\langle t_{\text{full}} \rangle / \langle t_{\text{surr}} \rangle$ | 667 | 162 | 25 |
| $x_{\text{max}}^{\text{p.m.}}$ | 234.03 | 544.96 | 1642.77 |
| $\epsilon_{\text{2nd,surr}}^{\text{p.m.}}$ | 8.5e−3 | 5.2e−3 | 1.8e−3 |
| $\alpha^{\text{p.m.}}$ | 0.9953 | 0.9958 | 0.9953 |
| $f_{\text{eff}}^{\text{p.m.}}$ | 1.93 | 0.29 | 0.02 |
| $x_{\text{max}}^{\text{med}}$ | 40.28 | 30.53 | 38.53 |
| $\epsilon_{\text{2nd,surr}}^{\text{med}}$ | 5.3e−2 | 8.5e−2 | 7.3e−2 |
| $\alpha^{\text{med}}$ | 0.9285 | 0.8204 | 0.4323 |
| $f_{\text{eff}}^{\text{med}}$ | 10.36 | 3.91 | 0.25 |

otherwise it shows a similar behaviour with the more extreme values of $x$ corresponding to small values of $w$.

The efficiencies of the initial unweighting step are also consistently lower than for the neutral gauge-boson channel. In particular for the process without external gluons, where $\langle t_{\text{full}} \rangle / \langle t_{\text{surr}} \rangle$ is 'only' 25, the factor of three between $\epsilon_{\text{full}}$ and $\epsilon_{\text{1st,surr}}$ might not be negligible. As expected given the larger values of $x_{\text{max}}^{\text{p.m.}}$ the corresponding efficiencies for the second unweighting step are all below 1 %, i.e. as low as 2 ‰ for $ud \to e^-\bar{\nu}_e duu\bar{d}$. However, for the median-reduced maximum the situation improves significantly, with $\epsilon_{\text{2nd,surr}}^{\text{med}}$ in the range of $5-8$ %. This efficiency improvement comes at the expense of the statistical power of the sample. While in the quantile approach the resulting $\alpha^{\text{p.m.}}$ factors are very close to unity, i.e. the effective sample size is larger than 99.5% of a true unit-weight sample, we observe more significant fractions of overweights with the median approach. This is true in particular for $dd \to e^-\bar{\nu}_e ggdu$ ($N_{\text{eff}} \approx 82\%N$) and $ud \to e^-\bar{\nu}_e duu\bar{d}$ ($N_{\text{eff}} \approx 43\%N$).

These performance measures are condensed into the resulting effective gain factor $f_{\text{eff}}$ according to Eq. (10). For the dominant $dg \to e^-\bar{\nu}_e gggu$ channel we find quite significant gains, even exceeding a factor of ten for the median approach. For the other channels the situation is different. For the all-fermion process surrogate unweighting needs *more* resources than the standard approach. This can be traced back to the relatively fast evaluation of the full weight, due to the simpler form of the matrix element, and the inferior performance of the NN in approximating the true event weights. However, in the global $W+4$ jets context, this channel contributes little to the total production rate and thus relatively few events need to be generated for such a channel. For the intermediate process, $dd \to e^-\bar{\nu}_e ggdu$, we find $f_{\text{eff}}^{\text{med}} \approx 4$. This speed-gain, however, also goes along with a more sizeable fraction of overweights, yielding $\alpha^{\text{med}} \approx 0.82$. We will therefore compare differential distributions for physical observable for this channel in the median approach next.

**Physics validation**

In Fig. 7 we present a comparison of physical distributions for the channel $dd \to e^-\bar{\nu}_e ggdu$ generated with and without the NN surrogate, employing $x_{\text{max}}^{\text{med}}$ in the second unweighting. We show results for (a) the transverse momentum of the charged boson, (b) the $k_t$ 4-jet resolution

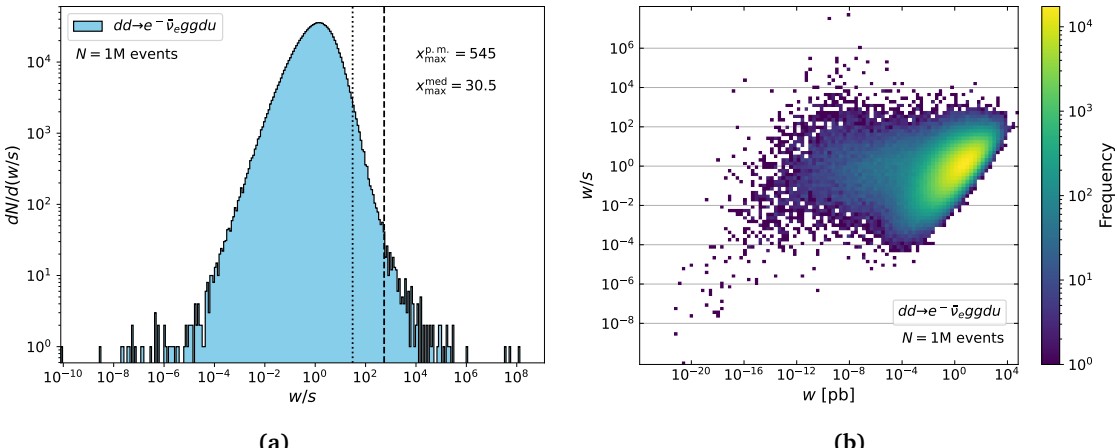

**(a)**               **(b)**

Figure 6: Distribution of weights using 1M test points generated with SHERPA for the process $dd \to e^- \bar{\nu}_e g g d u$ in proton–proton collisions at $\sqrt{s} = 13$ TeV. **(a)** One-dimensional histogram of the ratio $x = \frac{w}{s}$. The two vertical lines indicate the values of the reduced weight maxima $x_{\max}^{\text{p.m.}}$ (dashed) and $x_{\max}^{\text{med}}$ (dotted). **(b)** Two-dimensional histogram showing the relationship between the ratio $x = \frac{w}{s}$ and the true event weight $w$.

$d_{34}$, (c) the scalar sum of the four jet transverse momenta, $H_T$, and (d) the invariant mass of the two leading $p_T$ jets within the RIVET analyses MC_WINC, MC_WJETS, and MC_WKTSPLITTINGS.

For all four differential distributions we observe full statistical compatibility between the two samples of 1M events each. This further underlines that our surrogate-unweighting approach produces the exact target distribution. The considered observables all deeply probe the high-$p_T$ tails of phase space. In fact, the $p_T^W$ and $d_{34}$ distributions extend over five orders of magnitude in cross section. While for the given sample size of $N = 1$M we observe significant statistical fluctuations in the tails, these are fully consistent between standard and NN-surrogate generated samples. Even given $\alpha^{\text{med}} \approx 0.82$, corresponding to an effective sample size of $N_{\text{eff}} = 820$k, neither spikes or bumps are manifest in the nominal distributions, nor a significant increase in the statistical uncertainties for particular observable bins. And with a resulting gain factor of $f_{\text{eff}} \approx 4$, the surrogate method outperforms standard unweighting drastically. However, to some extent and as noted earlier, this statement depends on the post-processing procedures for the events. If the overall generation time of parton-level predictions is small compared to *e.g.* a full detector simulation, the standard unweighting might be preferable, at least for sub-channels with medium or low $f_{\text{eff}}$.

### 4.3.2   $t\bar{t}+3$ jets

Finally, we present results for processes belonging to the $t\bar{t}+3$ jets group. This probes the generalisation beyond the production of a single electroweak gauge boson in association with jets to a pure QCD process with massive particles. Even though the final state contains one particle less this process still poses a severe challenge. As top quarks carry colour charge there is a significant proliferation of Feynman diagrams when considering their jet-associated production. Despite these differences we employ the same neural-network architecture as before, adjusting the input-space dimensionality for the NN to 17, again utilising the three-momenta as input variables. We require three anti-$k_t$ jets with $R = 0.4$ and $p_{T,j} > 20$ GeV and do not impose phase-space cuts for the external top quarks. The latter are treated as on-shell in the matrix-element calculation, $p_t^2 = p_{\bar{t}}^2 = m_t^2$ with $m_t = 173.4$ GeV, and only decayed a posteriori to allow a more realistic definition of observables in the following physics validation.

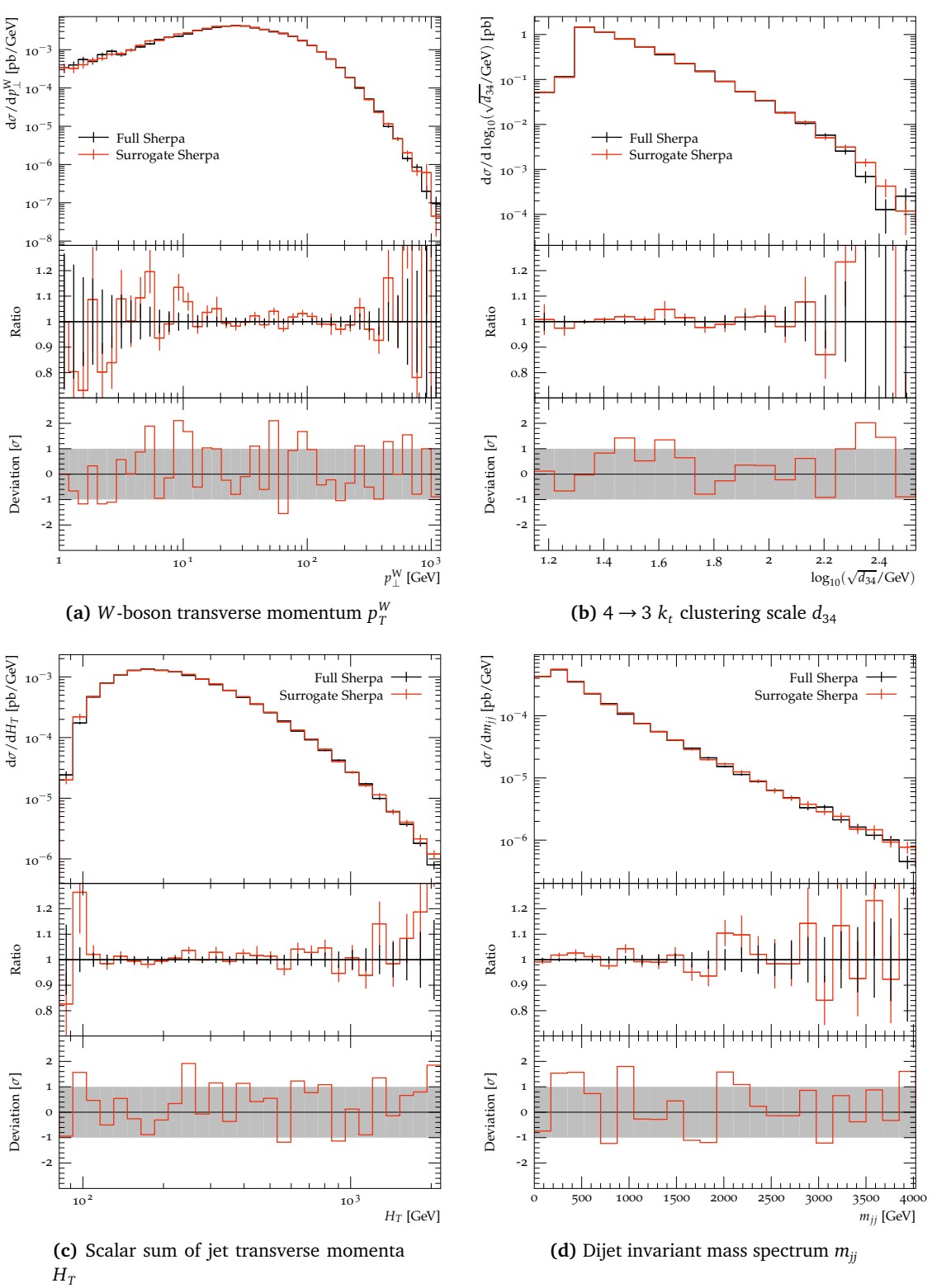

**(a)** $W$-boson transverse momentum $p_T^W$

**(b)** $4 \to 3$ $k_t$ clustering scale $d_{34}$

**(c)** Scalar sum of jet transverse momenta $H_T$

**(d)** Dijet invariant mass spectrum $m_{jj}$

Figure 7: Comparison of different differential distributions generated using SHERPA with (red) and without (black) an NN weight surrogate for the process $dd \to e^-\bar{\nu}_e g g d u$ in proton–proton collisions at $\sqrt{s} = 13\,\text{TeV}$.

In Tab. 5 we list the four considered partonic channels and their respective leading-order cross section in proton–proton collisions at $\sqrt{s} = 13\,\text{TeV}$.

Table 5: Selection of partonic channels contributing to $t\bar{t}+3$ jets production at the LHC and their corresponding leading-order production cross sections.

| process | cross section [pb] |
|---|---|
| $gg \to t\bar{t}ggg$ | 108.4(2) |
| $ug \to t\bar{t}ggu$ | 26.00(4) |
| $uu \to t\bar{t}guu$ | 3.733(8) |
| $u\bar{u} \to t\bar{t}gd\bar{d}$ | 0.01840(6) |

Clearly, under LHC conditions the all-gluon process has the largest production rate. In the second channel, *i.e.* $ug \to t\bar{t}ggu$ we instead consider an initial-state up-quark. Given that the QCD interaction does not change flavour, this parton species also appears in the final state. The third channel contains two up-quarks in the initial- and final state, corresponding to $t$-channel dominance in the top-quark production. The last considered process is $u\bar{u} \to t\bar{t}gd\bar{d}$, here top-quarks can be produced through $s$-channel gluons. Note, its production rate and correspondingly its contribution to an inclusive sample of unweighted events is significantly suppressed.

**Performance analysis**

In Table 6 we collect the performance measures for the surrogate-unweighting approach applied to the four top-quark productions channels. The reference unweighting efficiencies $\epsilon_{\text{full}}$ for standard unweighting with AMEGIC are typically higher than the ones found for $W+4$ jets before.

Table 6: Performance measures for partonic channels contributing to $t\bar{t}+3$ jets production at the LHC.

| | $gg \to t\bar{t}ggg$ | $ug \to t\bar{t}ggu$ | $uu \to t\bar{t}guu$ | $u\bar{u} \to t\bar{t}gd\bar{d}$ |
|---|---|---|---|---|
| $\epsilon_{\text{full}}$ | 1.1e−2 | 7.3e−3 | 6.8e−3 | 6.6e−4 |
| $\epsilon_{\text{1st,surr}}$ | 8.7e−3 | 5.8e−3 | 4.7e−3 | 3.6e−4 |
| $\langle t_{\text{full}} \rangle / \langle t_{\text{surr}} \rangle$ | 39312 | 2417 | 199 | 64 |
| $x_{\max}^{\text{p.m.}}$ | 52.03 | 32.52 | 69.76 | 326.19 |
| $\epsilon_{\text{2nd,surr}}^{\text{p.m.}}$ | 2.4e−2 | 3.8e−2 | 2.1e−2 | 5.6e−3 |
| $\alpha^{\text{p.m.}}$ | 0.9989 | 0.9984 | 0.9994 | 0.9981 |
| $f_{\text{eff}}^{\text{p.m.}}$ | 2.21 | 4.89 | 1.47 | 0.19 |
| $x_{\max}^{\text{med}}$ | 30.40 | 19.14 | 27.78 | 25.34 |
| $\epsilon_{\text{2nd,surr}}^{\text{med}}$ | 4.3e−2 | 6.4e−2 | 5.1e−2 | 7.1e−2 |
| $\alpha^{\text{med}}$ | 0.9983 | 0.9966 | 0.9943 | 0.9321 |
| $f_{\text{eff}}^{\text{med}}$ | 3.90 | 8.26 | 3.91 | 2.22 |

When comparing the evaluation times for the full event weights and the NN surrogate, quite significant speed-ups are found for $gg \to t\bar{t}ggg$ and $ug \to t\bar{t}ggu$. As before, a single evaluation of the surrogate weight takes about 0.12 ms. However, the weight calculation for

the all-gluon channel takes around 5 s, for $ug \rightarrow t\bar{t}ggu$ it is still around 0.3 s. Although we observe this high ratio of weight evaluation times, the effective gain factor $f_{\mathrm{eff}}$ is much smaller in the end and of the same order of magnitude than for the other processes. This can be mostly attributed to the relatively high unweighting efficiencies we start from. With a value of $\epsilon_{\mathrm{full}} = 1.1\mathrm{e}{-}2$ the process $gg \rightarrow t\bar{t}ggg$ has the highest unweighting efficiency of the examples considered here. According to Eq. 10, this clearly limits the possible gains. The reason for the high values of $\epsilon_{\mathrm{full}}$ is that this kind of multi-gluon channel is well-optimised in the integrator used by SHERPA.

The results obtained for $x^{\mathrm{p.m.}}$ and $x^{\mathrm{med}}$ are less spread out than for the $W{+}4$ jets processes. For $ug \rightarrow t\bar{t}ggu$ the NN performs best, with $x_{\mathrm{max}}^{\mathrm{p.m.}} \approx 33$ and $x_{\mathrm{max}}^{\mathrm{med}} \approx 19$. Only for $u\bar{u} \rightarrow t\bar{t}gd\bar{d}$ do we find an inferior performance with $x_{\mathrm{max}}^{\mathrm{p.m.}} > 300$. The values for the efficiency of the first unweighting step are comparable to what we found for the $Z{+}4$ jets channel, only for $u\bar{u} \rightarrow t\bar{t}gd\bar{d}$ it is significantly lower. Similar findings hold for $\epsilon_{\mathrm{2nd,surr}}^{\mathrm{p.m.}}$, which is lowest for $u\bar{u} \rightarrow t\bar{t}gd\bar{d}$. All effective sample size parameters are found to be larger than 0.99, with the exception of the $u\bar{u}$ process when using the median reduction method, where $\alpha^{\mathrm{med}} \approx 0.93$.

However, when using $x_{\mathrm{max}}^{\mathrm{med}}$ in the rejection sampling the effective gain factors are all higher than two, being largest for $ug \rightarrow t\bar{t}ggu$ with $f_{\mathrm{eff}}^{\mathrm{med}} \approx 8$. For the two computationally most expensive channels, that also feature the largest production rates, we obtain gains larger than two even with the per mille maximum reduction.

**Physics validation**

We close again by comparing predictions for physical observables, obtained with and without using the weight surrogate for the partonic channel $uu \rightarrow t\bar{t}guu$. Note, the on-shell top-quarks produced in the hard scattering get decayed with SHERPA's decay handler [6] prior to the final-state analysis. We here consider the semi-leptonic decay channel, *i.e.* $t\bar{t} \rightarrow l\nu_l q\bar{q}'b\bar{b}$ and employ the RIVET analysis MC_TTBAR.

In Fig. 8 we present exemplary results for (a) the invariant mass of hadronic $W$-boson candidates, (b) the $H_T$ distribution of all final-state jets, (c) the invariant mass of the hadronic top-quark candidates, and (d) the transverse momentum of the harder of the two final-state $b$-quark jets.

As before, we find full statistical agreement between the two samples for all considered observables. The rather fine binning of the invariant-mass distributions leads to larger statistical fluctuations for the given sample size of $N = 1\mathrm{M}$. However, as we will illustrate in Sec. 4.3.3 the deviations are in agreement with perfect statistical compatibility, *i.e.* both samples follow the same target distribution. Given $\alpha^{\mathrm{med}} = 0.9966$ we do not expect and in fact do not observe any visible effects from a reduced statistical accuracy of the sample produced with the surrogate approach.

### 4.3.3 Summary of physics validation for LHC processes

In addition to the selected observables for the three processes shown in the previous sections, we have performed a statistical compatibility analysis between the full and the surrogate setups based on 190 observables with almost 16,000 bins in total. The predictions are normalised in each observable for this analysis, to avoid a sensitivity to differences in the integrated cross section of each run, which would otherwise have to be accounted for as a correlation between different bins. As can be seen in Fig. 9, the deviations follow a normal distribution $\mathcal{N}(\mu, \sigma^2)$ with $\mu = 0$ and $\sigma = 1$, thereby validating our approach as faithful and unbiased.

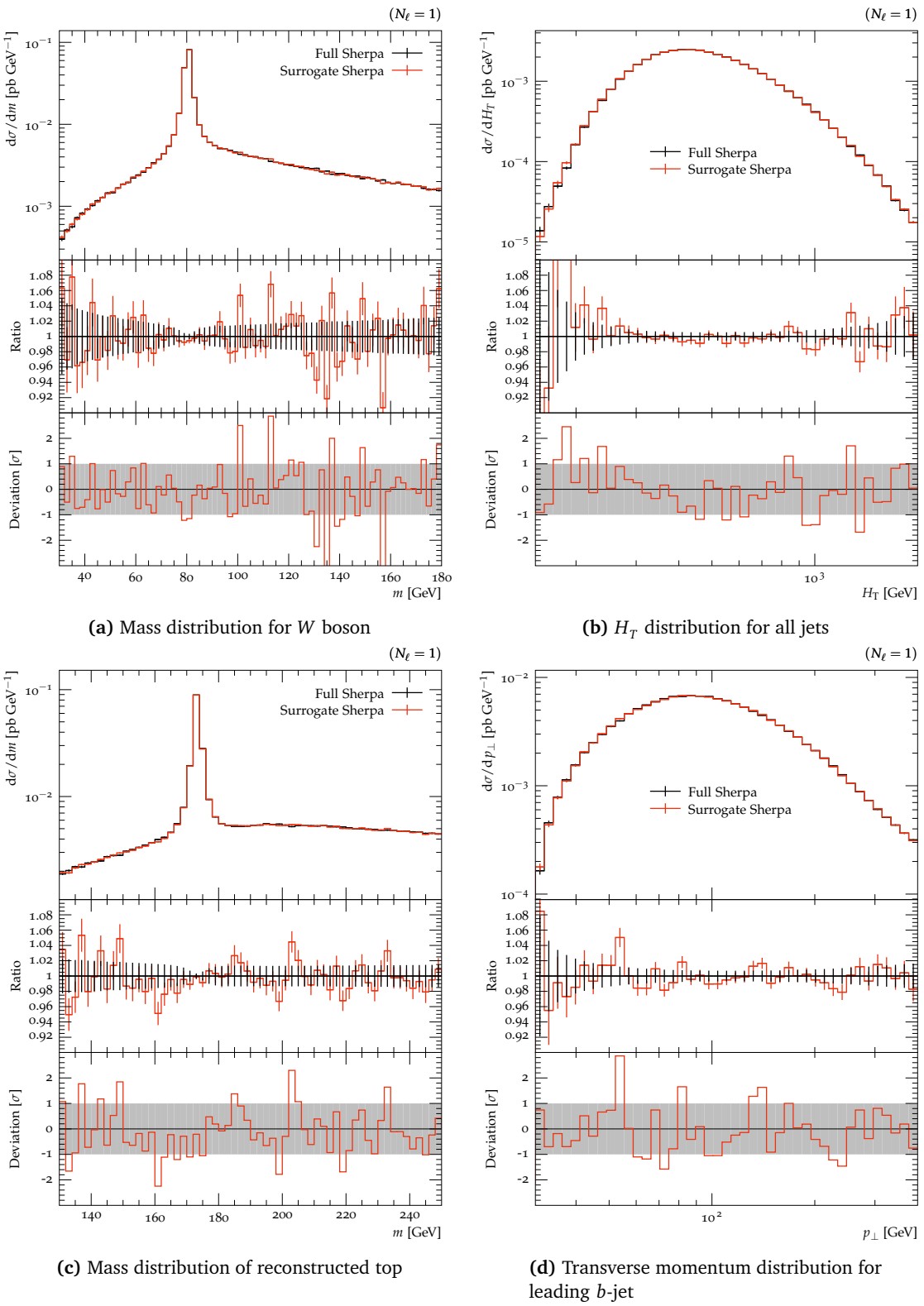

**(a)** Mass distribution for $W$ boson

**(b)** $H_T$ distribution for all jets

**(c)** Mass distribution of reconstructed top

**(d)** Transverse momentum distribution for leading $b$-jet

Figure 8: Comparison of different differential distributions generated using SHERPA with (red) and without (black) an NN weight surrogate for the process $uu \to t\bar{t}guu$ with subsequent leptonic top-quark decays in proton–proton collisions at $\sqrt{s} = 13$ TeV.

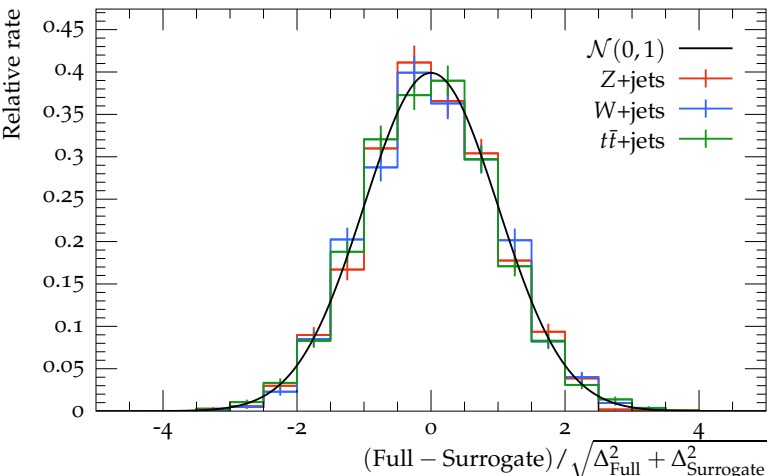

Figure 9: Distribution of deviations between the full and the surrogate approach in SHERPA from a comparison in almost 16,000 observable bins for all three processes.

## 5 Conclusions

Virtual particle collisions as simulated by Monte Carlo event generators play a central role in high-energy physics. Representing our best-knowledge theoretical expectations, they are used in the design and development of particle detectors for collider experiments, the planning and preparation of measurements, and, foremost, in the actual analysis and interpretation of real experimental data. To match actual measurements, particle-level virtual events need to be supplemented by a detailed simulation of the detector response. Given the calculational complexity and resource consumption of the detector emulation, ideally particle-level events with unit weight should be provided. However, the growing need in high-statistics simulations for a wide range of complex, high-multiplicity partonic scattering processes, including higher-order perturbative corrections, makes event unweighting a severe and very relevant computational challenge.

We have presented a novel two-staged unweighting algorithm that has the potential to significantly accelerate event unweighting. In an initial rejection-sampling step we employ a light-weight neural-network surrogate for the computationally expensive exact integrand, *i.e.* the matrix-element and phase-space weight. The mismatch of the surrogate and the true event weight is then corrected for in a second unweighting step. To protect against rare outliers in the true weight distribution as well as in the point-wise ratio of the true and the surrogate weight, we systematically reduce the respective numerically found maxima using a quantile or median approach, resulting in a partial overweighting of events. The relevant performance measures for the algorithm are the quality of the approximation, as well as the evaluation time per phase-space point, which can be combined into an effective per-event gain factor $f_{\text{eff}}$ with respect to conventional rejection sampling. This measure accounts for the reduced statistical power of the sample due to overweighting. It is used throughout this work to give a rigorous assessment of the effective improvement to be expected in various example processes. While the proposed unweighting algorithm has been developed in the context of collision-event simulations, it is in fact more general and can be used in other applications as well.

In Sec. 3 we have discussed the setup and training procedure used to approximate event weights with deep feedforward neural networks. As an initial test bed we have used a representative partonic channel contributing to tree-level $Z+4$ jets production at the LHC. We found

that our neural network is well capable of estimating the true event weights, thereby being more than 600 times faster.

In Sec. 4 we presented the practical implementation of the novel two-staged unweighting algorithm in the SHERPA event-generator framework. To further validate, benchmark and gauge the potential of the method, we applied it to high-multiplicity partonic channels contributing to $W+4$ jets and $t\bar{t}+3$ jets at the LHC. For the dominant partonic channels with sizeable cross sections and expensive matrix elements we found gain factors from using surrogate unweighting ranging from two up to ten. By comparing differential distributions of physical observables we were able to show that the proposed method indeed reproduces the correct target distribution. We were furthermore able to show that the partial overweighting of events, due to employing reduced maxima in the rejection sampling, barely affects the statistical accuracy and leaves no visible effect in physical distributions.

The unweighting algorithm presented here can also be applied in event generation beyond the leading order, where in parts of the phase space the event weights can become negative. While the proposed algorithm can take negative-valued weights into account, our SHERPA implementation is currently limited to tree-level matrix elements, where only positive weights appear. We leave the generalisation to NLO event generation and corresponding performance studies for future work. It will furthermore be interesting to apply our algorithm with alternative and potentially more powerful surrogate methods on the market, and evaluate their performance using the measures introduced in this work.

# Acknowledgements

We are grateful for fruitful discussions with Johannes Krause, Stefan Höche and Marek Schönherr, and Stephen Jiggins for reading the manuscript. We thank the Center for Information Services and High Performance Computing (ZIH) at TU Dresden for generous allocations of computing time.

**Funding information**   This work has received funding from the European Union's Horizon 2020 research and innovation programme as part of the Marie Skłodowska-Curie Innovative Training Network MCnetITN3 (grant agreement no. 722104). SS and TJ acknowledge support from BMBF (contracts 05H18MGCA1 and 05H21MGCAB). SS acknowledges funding by the Deutsche Forschungsgemeinschaft (DFG, German Research Foundation) - project number 456104544. FS's research was supported by the German Research Foundation (DFG) under grant No. SI 2009/1-1.

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
