# Peer review of "Accelerating Monte Carlo event generation -- rejection sampling using neural network event-weight estimates"

_SciPost Physics, doi:SciPost Phys. 12, 164 (2022)_

## Round 1 · Referee Report · Tilman Plehn (Referee 4) · 2022-2-12

Report

Thank you for all the clarifications and sorry for not getting one of the main points...

---

## Round 1 · Referee Report · Anonymous (Referee 1) · 2022-3-4

Dear Editor,

I am sorry to report that I am not convinced by the answers of the authors. Out of my four main points, two have not been answered convincingly, and another one has been fully ignored. It seems that the authors did not fully understand the points. Consequently, my referee report is conceptually the same as the first one and my recommendation of asking another revision as well. In this second report, I tried to be more specific to make sure that the authors do not miss the point.

1. Like in my previous report, I do agree with the authors that their algorithm holds for any numerical value of $w_{max}$. I further agree with them that too small values will lead to a poor performance. And I also agree that for a well-trained surrogate, this should be manageable.

   However, the authors claim that their statement concerning the effective sample size holds. I do not agree with that statement. For example, take the following case:

   $$\max_i(w_i) < w_{max} < \max_i(s_i), \text{ and } x_{max} > \max_i(x_i). \tag{1}$$

   In that case, standard one step un-weighting (using the same value of $w_{max}$) would not produce any over-weight. The two stage un-weighting would have some over-weight events (only from the first stage), but Eq. (6) claims that

   $$N_{eff} = N, \qquad \alpha = 1, \tag{2}$$

   as if all events were correctly un-weighted. Eq. (6) misses to include the over-weight related to the first un-weighting. The author should update that metric to include both source of overweight. Following their approach, I would suggest to use

   $$\alpha = \frac{1}{N} \frac{(\sum \tilde{w})^2}{\sum \tilde{w}^2}, \tag{3}$$

   which is the same as

   $$\alpha = \frac{1}{N} \frac{\left(\sum \max(1, s_i/w_{max}) \max(1, x_i/x_{max})\right)^2}{\sum \left(\max(1, s_i/w_{max}) \max(1, x_i/x_{max})\right)^2} \tag{4}$$

   or (if they want to compare to the standard over-weight allowed by AMEGIC algorithm):

   $$\alpha = \frac{\left(\sum \max(1, s_i/w_{max}) \max(1, x_i/x_{max})\right)^2}{\sum \left(\max(1, s_i/w_{max}) \max(1, x_i/x_{max})\right)^2} \frac{\sum \max(1, w_i/w_{max})^2}{\left(\sum \max(1, w_i/w_{max})\right)^2}. \tag{5}$$

   To be clear, those are suggestion of formula, I am open to any metric that does include both sources of overweight.

   As pointed by the authors in their answer, the additional Figure (3b) and (6b) indicates that $\max(s_i) < \max(w_i)$. Therefore, one can guess that the change of definition of $\alpha$ will not (or barely) impact the rest of the paper (if at all). Obviously, what is true in those examples does not mean it will be true for all cases, and the algorithm section should define a metric that takes into account all the sources of overweight (or at least all new sources of over-weights compared to the previous method).

   In their answer, the author claims that using $w_{max}$ rather than $s_{max}$, (which I understand as using the distribution of weight $w_i$ rather than $s_i$ for the determination of median/pm value)

they gain in performance at a cost of some additional over-weight. From the plots/tables of the paper, it is difficult to see why. If the authors have additional elements to support that statement, I would be interested to see them, but this is certainly not mandatory and not needed to be included in the paper.

2. I do believe that the authors fully miss this second point. So let me rephrase it. My remark here was why the author use Equation (5):

$$\tilde{w} = \max\left(1, \frac{x}{x_{\mathsf{max}}}\right) \cdot \max\left(1, \frac{s}{w_{\mathsf{max}}}\right) \tag{6}$$

and not the formula used in the literature for multiple stage un-weighting (for example see alpgen paper: hep-ph/0206293) –formula adapted to use authors convention–.

$$\tilde{w} = \max\left(1, \frac{x}{x_{\mathsf{max}}} \max\left(1, \frac{s}{w_{\mathsf{max}}}\right)\right) \tag{7}$$

As said in my previous report, it would be appropriate that the authors compare the two formula or comment why they do not think that the formula used in other code/context is relevant.

3. Here I failed to fully understand the answer of the authors, so let me be more specific in my issue/question.

If I look at their Eq. (12), the authors do include the Jacobian within the function learned by the Neural-Network (as they should do). However, that Jacobian depends not only on the four-momenta but also of the channel of integration used to generate that particular four-momenta. So I would expect that one need to train one surrogate for each channel of integration.

I would suggest that the author stress explicitly, if they train a single surrogate (my understanding of the paper) or one per channel of integration. If they use a single surrogate for all channels of integration, then it will be important to include a comment on the non-unicity of the Jacobian/weight for a given four-momenta and on the impact/limitation of that choice (or, in case, how that issue is avoided).

4. Sorry for the miss-understanding here. This fully answers my concern.

**minor points**

5. If you look at this paper: hep-ph/0006269, you will see that it presents the VEGAS algorithm as learning a surrogate function. In that paper, it is also presented that importance sampling is nothing else than a change of variable from a function on which you are able to generate events according to its density.

You can indeed see/present your algorithm as an extension of the hit-or-miss algorithm, but it also is an importance sampling method where the surrogate is learned by a Neural-Network and the density is obtained by hit or miss.

I do believe that mentioning, in the paper, that your method is an importance sampling method and that your method extends what VEGAS does, makes sense. But, as said, in the previous report, this is a minor point and I will not insist more on it any further.

6. Thanks for the modification.

7. I would suggest the authors to be more precise in their footnote. As they certainly know, a lot of non-expert are simply multiplying the weight by the generated cross-section, which is not correct in presence of over-weight. If the authors can provide the exact formula/method to use in presence of over-weights, this might help to limit the problem.

8. That's good to know.

9. Thanks to provide this reference. I would suggest to the authors to add that reference into the paper.

---

## Round 1 · Author Response

We would like to thank all the referees for their detailed and careful comments on our manuscript that helped us to further improve the presentation of our research results. We apologize for the delay in answering their queries, originating from the holiday season paired with the pandemics hitting us and our families.

In what follows we give answers to the various points raised and describe the corresponding
adjustments made to the paper. We address the different reviewers independently. However,
to avoid duplication we will sometimes refer to previous comments.

We have attached this reply letter to the pdf of the paper such that the mentioned auxiliary plots can be viewed.

Referee 1:

1.1 Our algorithm is in fact agnostic about the very choice of the maximum used in the
initial unweighting step, any particular choice of wmax or alternatively smax is compensated through an overweight according to Eq. (5). To this end, we are free to use either. Independent of the choice the statements about the effective sample size and the correctness of the algorithm hold, given that we properly account for the overweights. However, in case of smax>>wmax we would potentially face large corrections weights $\widetilde{w}$. That practically this is not a problem in our setup is illustrated for example in Fig. 4. To collect further evidence we have added two additional figures, Figs. 3b and 6b that show the ratios s/w in dependence of the true weight w. A corresponding discussion has been added to the text.

1.2 As stated before, we can work both with wmax or smax. Using wmax we gain performance in the unweighting, obviously for the price of overweights, that however, for a well trained surrogate are manageable, see 1.1.

1.3 Our approach is based on learning the integrand, i.e. squared matrix element and phase space density, from the physical three-momenta, cf. Sec. 3.1, which makes it transparent with respect to the used integrator. This would be different if we were to use random numbers as inputs for our NN. While details of AMEGIC do not play a role here, we certainly benefit from a good mapping of its multi-channel integrator, such that the spread of event weights, which our NN has to learn, is milder than that of the pure matrix element or phase space weights.
We have added a clarifying statement in Sec. 3.1 for these two points.

1.4 This is seemingly a misunderstanding, our validation is indeed for individual partonic channels, NOT their sum. In particular, we use the W+jets sub-channel which has an alpha=82%, which is relatively low compared to almost all other channels, but which would still benefit from the surrogate method (f_eff = 3.91 > 1). The only channel with a lower alpha is the last W+jets sub-channel. We could produce similar validation plots, but find it not very informative, because the surrogate method performs worse here than standard unweighting (f_eff = 0.25 < 1), and would thus simply not be applied in this sub-channel. However, for a practical (production-ready) implementation this is a point worth taking into closer consideration, to develop an algorithm to pre-determine these performance parameters and select sub-channels for which the NN unweighting is being activated.

1.5 We view our method as an extension of the hit-or-miss algorithm (rejection sampling,
von Neumann method etc) for unweighting, as defined in Algorithm 1 (3), rather than the
VEGAS sampling algorithm. The idea of the extension is then discussed in Algorithm 2 (4).
While VEGAS attempts to redistribute phase space variables, i.e. random numbers, we
here learn a surrogate for the full matrix element times phase space weight, to perform
unweighting. We are not aware of this being used in the literature before, but are happy
to refer to previous work suggesting such a two-step unweighting, if brought to our
attention.

1.6 This is a very important point that is being raised by the referee, and in fact one
that we have taken into rigorous consideration by defining a measure f_eff, which
takes the reduced statistical power of such a sample into account in a sophisticated way.
We mention this measure in the conclusions explicitly and have now added a sentence to
stress that exactly this compensation is included in f_eff.

1.7 Indeed we use 'weights' with different dimensions. While the event weights w and the
surrogates s are dimensionful, any ratios, for example overweights $\widetilde{w}$ are not.
As conventionally done, for unweighted event samples these need to be normalized in the end
to the generated cross section.

For clarification we have added a footnote on p5.

1.8 The NN training is done only once, and in practice can re-use the phase space points and
matrix element evaluations during the initial integration phase. Thus the CPU cost for this
is rather negligible compared to the final event generation.

1.9 Alternative fitting methods have been studied in a MSc thesis at TU Dresden, see

https://iktp.tu-dresden.de/IKTP/pub/15/masterthesis-2015-10-johannes-krause.pdf

When comparing between random forests (based on decision trees) and neural networks,
the effective timing for event generation has been dominantly in favor of neural networks,
as can be seen in the Figure below (Fig. 5.2 in the thesis). With these findings we did not further pursue these alternative approaches.

Referee 2:

2.1 We have added with Figs. 3b and 6b two-dimensional plots of w/s vs. w and amended the
text accordingly. The results clearly support and reassure that the vast majority of points have
w/s\approx 1 and rare outliers with s>>w indeed appear for rather small values of w only.

See also reply 1.1 above.

2.2 As mentioned in reply 1.3 we do not specifically cater to a given integrator method, but
are agnostic to that, i.e. the differential and Jacobian in Eq. (12) can come from an
arbitrary integrator. We added a statement to Sec. 3.1 to clarify that matter and hint at the interplay with the multi-channel method.

2.3 The ratios in Fig. 5, 7, 8 are already at the few-\% level and we do not see any hint of
significant deviations or biases that could play a phenomenological role. Accordingly we consider the given statistics sufficient for the given purpose.

To further comment on the mentioned weaknesses:

To address the dependence of the results/performance on the maximal weight we have
selected two different approaches for defining the maximum and compared them to get an
assessment of this new handle on speed.

The observed performance pattern for the different subprocesses is caused by a different
unweighting performance of the Sherpa integrators for different partonic channels,
i.e. for the multi-gluon channel the vanilla integrators are already optimized well
and the unweighting efficiency is relatively high to start with (cf. end of Sec. 2.1).
The novel ML approach can thus “only” improve this by a factor of 4.
A brief comment has been added at the end of Sec. 4.3.2.

Referee 3:

Weaknesses:

3.1 While there have recently been publications on NN surrogates for NLO matrix elements,
to the best of our knowledge no algorithm to combine this with associated phase space weights
and their using this for event unweighting, thereby guaranteeing the exact target distribution,
has been presented so far. Our algorithm can straightforwardly be applied for NLO event
generation, possibly using other peoples results/findings for optimal NN representations of
NLO matrix elements. In fact this is high on our to-do list.

3.2 See our replies 1.1 and 1.2, our method guarantees that we sample the right target
distribution, by properly accounting for potential overweights.

Changes:

Indeed the x-axis label was wrong, thanks for spotting. We have corrected this.

Concerning the quoted evaluation time in the introduction, we would like to avoid too many
technical details here. Full details for any timing can be found in Ref. [25] that
is cited at this point.

Referee 4 (Tilman Plehn):

4.1 This seems to be a misunderstanding: We are never ignoring any weight induced by
our algorithm or use any approximation. Our distribution is faithful by construction,
i.e. the inaccuracy of the surrogates is fully corrected for in a second unweighting
step. The whole point of our method is to find a computationally cheap estimate for the
combined matrix element and phase space weight such that the number of target function
evaluations in the unweighting procedure get reduced.

4.2 The hyperparameters for the two methods of defining the maximal weight just steer the
efficiency of the event generation. The latter is discussed in more detail in Sec. 3.2.
See also our reply to referee 2.

4.3 We have added a new figure in Fig. 3b and Fig. 6b which shows that the tails
typically affect events with a low differential cross section event weight.
In other words, the majority of the events will be generated efficiently, and only
rare events are less efficient in the second unweighting.

4.4 As to where the extreme x events lie in terms of phase-space observables: This
depends on the process and on the mapping used by the integrator. For the Z+jets process
corresponding to Fig. 3, these events show up in the high jet pT tails, as can be seen
in the figures below. While these are very unlikely phase-space configurations according
to the cross section, Sherpa severely over samples these regions leading to very small
weights. For other processes we would expect something similar: The events with
small/large x values will show up in regions that are oversampled by the integrator
which typically happens in the low cross-section tails.

4.5 To understand why the surrogate struggles most at small event weights, it is
helpful to look at the distribution of the mean squared error shown below, as this is
used as the loss function for training the NN. One observes that in contrast to the
relative deviations appearing in the x-values, the MSE is largest for high weight
values. This is to be expected as the MSE loss penalizes deviations at larger values
(quadratically) more than at smaller values.

4.6 We also observe that the largest and especially the smallest $x$-values appear
for events with very high momentum fractions of the initial-state partons. We assume
that this reflects the fact that high momentum fractions are necessary to reach the
high $p_T$ tails where the smallest weights can be found.

4.7 We prefer the algorithmic presentation, which also makes it easier to see the
differences between the algorithms.

4.8 We are attaching the Rjj distributions between all combinations of the leading 3 jets here. We have not added them to the manuscript, since we do not think they add much information beyond the existing plots in Fig. 5. Our algorithm does not single out any specific observable.

4.9 The evaluation time is related to the complexity of the underlying matrix element,
so yes, the number of Feynman diagrams, number of helicities etc are key figures here.

4.10 Training on a combination of partonic channel is certainly a viable approach,
but the resulting generation efficiency would have to be tested. However, at present
the training of the model is actually not a limiting factor, so there is no strong need
for a combined training. This might change when going NLO and using more complex network
architectures.

4.11 See reply 4.8, we are never ignoring any weight induced by our algorithm. Our
distributions in Fig. 5, 7, 8 are thus faithful by construction, i.e. the
inaccuracy of the surrogates is fully corrected for in a second unweighting step.
The statistical agreement would look identical with any surrogate model. The
difference will only show up in tables 2, 4, 6 as a reduced overall efficiency factor f_eff.

4.12 As stated before, going NLO is high up on our agenda. Stay tuned ;)

4.13 We have added the additional reference in the introduction, as well as 2110.13632,
sorry for the omission in the first place.

Again would like to thank the referees for their careful assessments of our paper. With our replies their questions and the corresponding changes to the manuscript we hope that our paper meets all expectations to be published in SciPost.

All the best,
the authors

---

## Round 1 · List of Changes

see above

---

## Round 2 · Referee Report · Tilman Plehn (Referee 4) · 2022-3-28

Report

Thank you for addressing all my comments, and sorry for the misunderstanding. I think the paper is seriously cool and a significant step in ML-enhanced event generation, so let's publish it!

---

## Round 2 · Referee Report · Anonymous (Referee 1) · 2022-4-7

Report

I thank the authors for their careful consideration of my comments and the associated change to the text, which clarify and improves the global quality of the paper. This fully answer all my concerns, and therefore I'm recommending to publish the paper in his current form.

I would like to apologize to the authors since I was clearly missing a point on the multichannel. Thanks a lot for the clarification.

For the record, I would like to comment on the new validation plot made by the author in their last answer. I have done the same toy example as them and do not observe any bias. But I do not think that investigate our disagreement is relevant for the publication of the paper.

---

## Round 2 · Author Response

We would like to thank the referee for the detailed second report and for re-iterating details on several points to clarify misunderstandings in our reading of them. This is very much appreciated! We have carefully revisited the points raised and compile our responses below, thereby following the ordering of the referee. We have accordingly adjusted, extended and clarified the text in the paper, as detailed below.

  1. We thank the referee for spotting this mistake of ours. It is correctly pointed out that our Eq. (6) does not include the overweights from the first unweighting stage. While this does not affect our algorithm it has obvious consequences for the employed performance measures. We have corrected Eq. (6) to what the referee has given as Eq. (3) in his/her report. In fact, the former version of Eq. (6) was actually a typo. In our implementation of the performance measures computation we had used the proper form including all overweights in the determination of the α factors already. Accordingly, none of the quoted results needed to be corrected.

  2. The reference to the Alpgen paper is helpful and we thank the referee for pointing us to it. We think it has some similarities with our method which we were not aware of. Accordingly, we added this reference to our revised paper. However, our overweight treatment has been developed independently and our Eq. (5) is the approach we consider appropriate for the case at hands. The fact that there is no indication of mishandling of the overweights in our examples (see toy example Fig. 1, deviations plot Fig. 9) convinces us of the correctness of our equation. It is, however, not equivalent to the equation(s) suggested by the referee. We have also found no counterpart for the referee’s equation in the given reference. However, to study this further, we applied both equations to a simple toy example. The results can be seen in the attached plot (see resubmitted paper, at the very end). We find that the suggested formula does not reproduce the target function, implying that it handles some of the overweights wrongly. These can be attributed to the case where x<x_max and s>w_max. Our treatment correctly accounts for all overweights. Possibly there could be a typo in the equation suggested by the referee. In any case, we see no reason to change our expression given that it produces correct results.

  3. To more clearly illustrate the interplay/dependence of our method with the used sampling technique for generating events, we have extended the discussion in Sec. 2 now also briefly introducing importance sampling and the multi-channel method, see new Eqs. (4)-(6). This is then picked up in the discussion of our actual deliverable, i.e. fully differential cross section integrals, in Sec. 3. We have significantly extended the discussion of Eq. (15) and elaborate on possible alternative treatments for a multi-channel sampler. The case we present in our paper indeed uses a single NN with the external particles’ three-momenta as input variables, that are generated by a (multi-channel) probability density specific to Amegic and the considered process. Our network thus effectively learns the ratio f/g (with g the total mapping function, i.e. the sum of all channels). However, as we are not using random numbers as the input variables, that have channel specific mappings to momenta, we do not need to keep track of the individual channels for example through channel specific NN surrogates or via the random number used to select the channel as further input variable.

  4. Clearly, as we point out in our toy example already, the surrogate can be from whatever source, including a VEGAS grid or any other importance sampling density. This is also touched upon in the intro to Sec. 3. However, we here concentrate and explore the potential of NNs that we believe have particular promising capabilities.

  5. We have added a sentence and an equation to the relevant footnote to stress the fact that non-expert users have to rely on the generated cross section as calculated by the MC program, which should contain the correct normalisation for the given set of events that have been generated. We prefer to not single out the overweight case with further equations for sigma_gen, since this would have to include not only overweight events faithfully but also correctly include N_trials from the unweighting and from potential rejections in ME+PS merging, (negative) weights from the NLO+PS matching procedure, phase space biasing weights, and other advanced features of modern MC programs.

  6. We have made the thesis available on CDS and included an explicit reference, Ref. [91].

Again we would like to thank the referee for insisting, which has helped us to make the manuscript significantly clearer! We hope that the paper in its present form qualifies for publication in SciPost Physics.

---

## Round 2 · List of Changes

- extended discussion at the introduction to Sec. 2
- former Eq. (6), now Eq. (8), has been corrected
- extended discussion in Sec. 3.1
-updated footnote 3, concerning sample normalisation

---

## Editorial Decision

published